# Reliable Models via Responsiveness Verification

## Abstract

Many safety failures in machine learning arise when models are used to assign predictions to people – often in settings like lending, hiring, or content moderation – without accounting for how individuals can change their inputs under realistic constraints and imperfect data. In this work, we introduce a formal validation procedure for the responsiveness of predictions with respect to interventions on their features. Our procedure frames responsiveness as a type of sensitivity analysis in which practitioners control a set of changes by specifying constraints over interventions and distributions over downstream effects, allowing uncertainty from biased, truncated, or missing data to be made explicit. We describe how to estimate responsiveness for the predictions of any model and any dataset using only black-box access, and how to use these estimates to support tasks such as falsification and failure probability estimation. We develop algorithms that construct these estimates by generating a uniform sample of reachable points, and demonstrate how they can promote safety in real-world applications such as recidivism prediction, organ transplant prioritization, and content moderation.

## 1 Introduction

Many of the pressing safety issues with machine learning arise in cases where model predictions affect *people* [54] – be it to approve loans [27], screen job applicants [6, 49], prioritize organ transplants [3, 7, 43], or moderate posts on online platforms [20, 22]. In such applications, we fit models that use features about individuals for predictions but cannot account for the changes in the predictions if the features are *intervened* upon. As a result, we routinely deploy models whose predictions are either not *responsive* to the actions of their decision subjects, or are overly *responsive*.

When the models are under-responsive, they can preclude access to loans, jobs, or healthcare [57]. In lending, for example, models can preclude credit access by assigning fixed predictions that applicants cannot change [33]. In healthcare, models can prolong wait times for organ transplants by assigning predictions on the basis of patient characteristics such as age [7, 43]. When models are overly responsive, they exhibit unfairness [34], or are susceptible to gaming [25]. For instance, in content moderation, models can promote the proliferation of misinformation by allowing malicious actors to evade moderation at scale [1, 50].

A central challenge in addressing these issues is measuring the responsiveness of predictions – i.e., by how much the output of a model can change over a space of plausible feature vectors. Measuring this quantity in practice hinges on our ability to effectively specify the set of plausible feature changes. In applications where features encode semantically meaningful characteristics, this set must adhere to non-trivial constraints on both the plausible interventions and their downstream effects. Choosing a set that is too small can underestimate responsiveness by overlooking viable interventions, whereas choosing a set that is too large can overestimate responsiveness by considering unrealistic changes that no individual could enact.

Submitted to 39th Conference on Neural Information Processing Systems (NeurIPS 2025). Do not distribute.

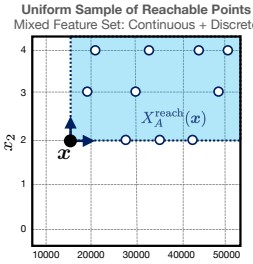 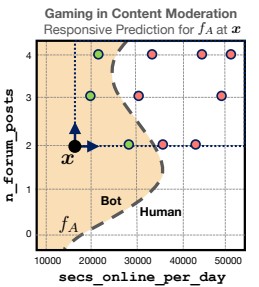 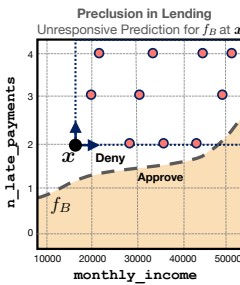

**Figure 1:** Responsiveness verification with reachable sets. **Left**: Given an instance $\boldsymbol{x}$, we generate a *uniform* sample of reachable points $X_A^{\text{reach}}(\boldsymbol{x})$, i.e., feature vectors that can be reached following an intervention on $\boldsymbol{x}$. Here, $x_1 \in [10000, 60000]$ and $x_2 \in \{0, \ldots, 5\}$ are monotonically increasing features. **Middle**: We use $X_A^{\text{reach}}(\boldsymbol{x})$ to determine the vulnerability to *gaming* in a bot detection task; accounts that are flagged as bots should not be able to readily change their prediction without human review [20, 22]. Here, $f_A$ is susceptible to gaming at $\boldsymbol{x}$. **Right**: We use $X_A^{\text{reach}}(\boldsymbol{x})$ to test for *preclusion* in a lending task by testing if an applicant can change their prediction to be approved [33] and see that $\boldsymbol{x}$ is precluded from access under $f_B$.

In this work, we present a formal procedure to validate models by measuring the responsiveness of their predictions under realistic constraints on interventions. Our aim is to provide practical validation tools that (i) remain applicable with only black-box model access, (ii) surface safety failures with examples, and (iii) explicitly account for strategic behavior and imperfect data. To this end, we develop machinery that can support formal validation tasks such as *falsification* and *failure probability estimation* [31]. More broadly, we seek to overcome barriers to adoption of models by developing a validation framework that is widely applicable, and by demonstrating its capabilities. Our contributions include:

1. We introduce a formal procedure to estimate and test the responsiveness of predictions for models with semantically meaningful features. Our procedure can specify fine-grained constraints on interventions and their downstream effects. This allows practitioners to reveal failures that affect individual or system-wide safety, estimate their prevalence, and pair each failure with examples.
2. We develop algorithms to estimate the responsiveness of predictions for any model and any dataset using only query access. Our methods generate a uniform sample of reachable points from a non-convex polytope over discrete and continuous features, and benefit from simple theoretical guarantees that can guide practical decisions in test design.
3. We demonstrate how our machinery can reliably detect inadvertent failures in responsiveness in model development or deployment. We illustrate this through real-world applications in recidivism prediction, content moderation, and organ transplant prioritization.
4. We provide a Python library to estimate and test responsiveness at this anonymized repository.

**Full Version and Supplementary Materials** In the supplement we include: (1). Related work (2). A description of our sampling algorithm and pseudocode (3). Additional experiments

## 2 Framework

We describe a formal validation procedure to test if a machine learning model assigns predictions that are unsafe as a result on interventions on its features. We consider a task where we are given *black-box access* to a model $f : \mathcal{X} \to \mathcal{Y}$ to predict an outcome $y \in \mathcal{Y}$ from a set of $d$ *features* $\boldsymbol{x} = [x_1, \ldots, x_d] \in \mathcal{X}$. We assume that features are *semantically meaningful*, e.g., features that encode meaningful characteristics for the task at hand like `income` and `employment_status` as opposed to generic features such as pixel intensities or token embeddings.

We consider a procedure where we validate a model by testing its predictions over a *target population*. We assume the target population covers all points $\boldsymbol{x} \in \mathcal{X}$, or a subset of instances we can identify from their features and/or predictions (e.g., all instances with features $\boldsymbol{x}$ such that $f(\boldsymbol{x}) = 0$). We test if a model assigns *unsafe* predictions by measuring the *responsiveness* of predictions:

> **Definition 1.** Given an instance with features $\boldsymbol{x} \in \mathcal{X}$ and a model $f : \mathcal{X} \to \mathcal{Y}$, the *responsiveness*

of its prediction $f(\boldsymbol{x})$ is the proportion of interventions that lead to a target prediction:

$$\rho(\boldsymbol{x}; f, X_A^{\mathrm{reach}}, \hat{Y}^{\mathrm{reach}}) = \Pr\left(f(\boldsymbol{x}') \in \hat{Y}_{\boldsymbol{x}}^{\mathrm{reach}} \mid \boldsymbol{x}' \in X_A^{\mathrm{reach}}(\boldsymbol{x})\right),$$

Here:

- $X_A^{\mathrm{reach}}(\boldsymbol{x}) \subset \mathcal{X}$ is a set of reachable points, determined by the types of interventions we allow. We denote the set of all possible interventions at a point $\boldsymbol{x}$ as $A(\boldsymbol{x})$, and refer to it as the *intervention set*. We assume that includes a null action $\boldsymbol{0}$.
- $\hat{Y}_{\boldsymbol{x}}^{\mathrm{reach}} \subseteq \mathcal{Y}$ is a *target prediction*, which can represent a single value in a binary classification task (e.g., $\hat{Y}_{\boldsymbol{x}}^{\mathrm{reach}} = \{1\}$), a set of values in a multiclass classification task (e.g., $\hat{Y}_{\boldsymbol{x}}^{\mathrm{reach}} = \{\texttt{spam}, \texttt{hate\_speech}\}$ in content moderation), or an interval in a regression task (e.g., $[700, 850]$ in credit scoring). We write $\hat{Y}_{\boldsymbol{x}}^{\mathrm{reach}}$ to allow the target prediction to change based on $\boldsymbol{x}$.

In what follows, we write $\rho(\boldsymbol{x})$ when the model, target prediction and reachable set are clear from context. We can adapt our framework to various formal validation tasks:

- *Preclusion*: Consider testing if a loan approval model $f : \mathcal{X} \to \{0, 1\}$ assigns "fixed" predictions that preclude credit access [9, 33]. Here, the target population covers all denied applicants i.e., $\{\boldsymbol{x} : f(\boldsymbol{x}) = 0\}$. Given a point $\boldsymbol{x} \in \mathcal{X}$, we estimate the responsiveness of each prediction $\hat{\rho}(\boldsymbol{x})$ to see if there exists some interventions that could overturn the current prediction to $\hat{Y}_{\boldsymbol{x}}^{\mathrm{reach}} = \{1\}$. Given the estimate, we would test if $\rho(\boldsymbol{x}) > 0$ and claim that the model precludes access if we cannot refute the claim that $\rho(\boldsymbol{x}) = 0$.
- *Gaming*: In a content moderation task where we use a model to detect bot accounts, we may wish to test if bot accounts can alter their features to pass as a human end-user. In this case, we would estimate the responsiveness of an account who is predicted as a bot. Contrary to lending, we could have a toleration threshold $\varepsilon$ and raise a safety violation if $\rho(\boldsymbol{x}) > \varepsilon$. We can also estimate responsiveness of individual predictions to characterize each point or compute aggregate responsiveness statistics to describe the model (i.e., mean responsiveness).
- *Unaffordability*: In an insurance task, where we use a regression model to determine a monthly insurance premium, we may wish to test that the premium remains affordable for each instance even if we diagnose a pre-existing condition [10]. In this case, our test population would represent all instances $\boldsymbol{x} \in \mathcal{X}$ and our target prediction $\hat{Y}_{\boldsymbol{x}}^{\mathrm{reach}} \subset \mathbb{R}$ could change based on their income.

**Interventions and Downstream Effects** The reliability of these procedures depends on how we specify the set of reachable points. Consider estimating if a model could be gamed by measuring the responsiveness of a prediction with respect to all interventions over $\|\boldsymbol{a}\|_p \leq \delta$. In this case, our claims and estimates depend on how we set $\delta$: small values may lead to blindspots while large values may lead to false alarms [see 30, for a discussion]. In tasks with semantically meaningful features, we can rarely mitigate these issues by setting $\delta$ because this practice provides no control over actionability. For example, a decision subject may be unable to change some features, which leads us to consider infeasible interventions. Alternatively, deliberate interventions could induce changes on others features and probabilistic changes on others (e.g., taking a medication may alter a patient's blood pressure). We would overlook these effects if we only consider constraints that pertain to a single feature – immutability, bounds or monotonicity.

We consider a general model that distinguishes *interventions* from *downstream effects*.

**Definition 2.** *Given an instance $\boldsymbol{x}$, we assume that an intervention changes its features as:*

$$\boldsymbol{x}' = \boldsymbol{x} + \boldsymbol{a} + \boldsymbol{r},$$

*Here, $\boldsymbol{a} \in \mathbb{R}^d$ captures an* intervention *– i.e., a deliberate action performed by an individual. In turn, $\boldsymbol{r} \in \mathbb{R}^d$ specifies* downstream effects *that stem from the intervention.*

We follow [29] and call a feature *actionable* if it can be directly changed by a decision subject, and *inactionable* otherwise. Our model allows practitioners to specify intervention set $\boldsymbol{x}$, and a conditional probability distribution to specify downstream effects $\mathbb{P}_{\boldsymbol{x}, \boldsymbol{a}}(\boldsymbol{r})$. This representation allows us to specify different classes of downstream effects:

- *Fixed Effects* [9], where interventions induce deterministic changes on features due to feature encoding or deterministic causal effects. In a lending task, we can express a deterministic down-

stream effect such as `n_monthly_payments` $= 12 \to 24$ will increase age by 1 as $\mathbb{P}_{\boldsymbol{x},\boldsymbol{a}}(r_k) = 1$ if $r_k = \lfloor \frac{a_j}{12} \rfloor$ where $j$ and $k$ denote `n_monthly_payments` and `age`, respectively.

- *Random Effects*, which capture random effects in feature values that arise independently of the intervention – e.g., due to noisy measurements or natural variability across repeated predictions. For instance, in a risk scoring system for organ transplant allocation, a patient's score could be responsive to normal physiological fluctuations. As an example, bilirubin levels – used in liver-disease risk scores such as MELD [28] – may vary across lab tests for the same person. In such cases, we could have that $\mathbb{P}_{\boldsymbol{x},\boldsymbol{a}}(\boldsymbol{r})$ is independent of $\boldsymbol{x}, \boldsymbol{a}$, and $\mathbb{P}_{\boldsymbol{x},\boldsymbol{a}}(r_{\text{bilirubin}}) = \mathcal{N}(0 \text{ mg/dL}, 5 \text{ (mg/dL)}^2)$.

- *Causal Effects*, where downstream effects are sampled from a probability distribution that we obtain from applying an intervention on a structural causal model [see, e.g., 29, 47]. Consider a car insurance risk scoring task, where the feature `annual_distance` representing distance driven causally depends on whether the driver works remotely, indicated by an actionable binary `remote_work` feature. We can model this as:

$$\mathbb{P}_{\boldsymbol{x},\boldsymbol{a}}(r_{\text{annual\_distance}}) = \begin{cases} \mathcal{N}(0 \text{ km}, 1000 \text{ km}^2), & a_{\text{remote\_work}} = 0 \\ \mathcal{N}(6000 \text{ km}, 1000 \text{ km}^2), & a_{\text{remote\_work}} = 1 \end{cases}$$

| Class | Example | Features | Constraint |
|---|---|---|---|
| Immutability | `content_created_at` should not change | $x_j = $ `content_created_at` | $a_j = 0$ |
| Monotonicity | `patient_age` can only increase | $x_j = $ `patient_age` | $a_j \geq 0$ |
| Integrality | `n_visits` must be positive integer | $x_j = $ `n_visits` | $a_j \in \mathbb{Z} \cap [0 - x_j, 52 - x_j]$ |
| Feature Encoding | preserve one-hot encoding of `patient_type` $\in \{$`In`, `Out`$\}$ | $x_j = $ `patient_type_in` 
 $x_k = $ `patient_type_out` | $a_j + x_j \in \{0,1\} \quad x_k + a_k \in \{0,1\}$ 
 $a_j + x_j + a_k + x_k = 1 \quad a_j + x_j \geq a_k + x_k$ |
| Missing Values | if `no_posts` $= $ `TRUE` 
 then `num_posts` $= 0$ 
 else `num_posts` $\geq 0$ | $x_j = $ `no_posts` 
 $x_k = $ `num_posts` | $a_j + x_j \in \{0,1\} \quad a_k + x_k \in [0, 10^7]$ 
 $x_j \cdot x_k = 0 \quad x_k \geq 1 - x_j$ |

**Table 1:** Examples of constraints on interventions. Each constraint can be expressed in natural language and embedded into an optimization problem using standard techniques in mathematical programming [see 60].

**Discussion** In many of the use cases above, we can promote safety by detecting predictions that are unsafe with respect to a *minimal response model*. In a preclusion detection task, for example, a minimal model would capture constraints and distributions that are indisputable – e.g., interventions must ensure the integrity of feature encoding, and distributions must induce deterministic downstream effects. If we are able to detect instances of preclusion even under this minimal model, then it would imply that preclusion is likely to arise under any other realistic constraints as well.

## 3 Estimating Responsiveness

In this section, we describe a general framework to verify the responsiveness of predictions. Consider a probability distribution over the reachable points in $X_A^{\text{reach}}(\boldsymbol{x})$ – i.e., $\boldsymbol{x} + \boldsymbol{a} + \boldsymbol{r} = \boldsymbol{x}' \sim \mathbb{P}_{\boldsymbol{x}}^{\text{reach}}$, where we set $\boldsymbol{a} \sim \text{Uniform}[A(\boldsymbol{x})]$ to ensure coverage over the entire space of interventions, and $\boldsymbol{r} \sim \mathbb{P}_{\boldsymbol{x},\boldsymbol{a}}(\cdot)$ according to our model of downstream effects. Given an instance $\boldsymbol{x}$, we can compute its responsiveness using i.i.d. samples from this distribution:

$$\rho(\boldsymbol{x}) = \mathbb{E}_{\boldsymbol{x}' \sim \mathbb{P}_{\boldsymbol{x}}^{\text{reach}}} [\mathbb{I}[f(\boldsymbol{x}') \in \hat{Y}_{\boldsymbol{x}}^{\text{reach}}]]$$

Given a model $f$, we estimate this quantity from $n$ i.i.d. sampled points $\hat{X}_n \sim (\mathbb{P}_{\boldsymbol{x}}^{\text{reach}})^n$ as:

$$\hat{\rho}_n := \frac{1}{n} |\{\boldsymbol{x}' \in \hat{X}_n : f(\boldsymbol{x}') \in \hat{Y}_{\boldsymbol{x}}^{\text{reach}}\}| = \hat{S}_n / n$$

This approach has several benefits:

- We can estimate the responsiveness of predictions for any model. Our approach does not depend on model type and only requires black-box query access.
- It yields simple but reliable statistical guarantees with respect to sample size $n$ and a desired confidence level. This is a result of building our estimates from i.i.d. samples, allowing $\hat{S}_n$ to be a binomially distributed random variable.
- We can extract a set of points $\mathcal{X}_{\text{unsafe}} \subseteq \mathcal{X}$ that induce the failure mode and analyze them to facilitate debugging (e.g., identifying problematic features).

In what follows, we describe how to estimate responsiveness from a sample of reachable points, then describe how to generate uniform samples of reachable points across interventions in practice.

**Procedures and Guarantees**    We show how to use reachable sets to certify responsiveness of a prediction.

---

**Proposition 3** (Estimation).    *Consider estimating the responsiveness of the prediction at $\boldsymbol{x}$ from a model, $f$. Given an estimate $\hat{\rho}_n$ from $n$ reachable points $\hat{X}_n$ and confidence parameter $\alpha \in (0,1)$, denote the confidence interval as:*

$$\hat{C}_\alpha(n, \hat{\rho}_n) := [\mathsf{B}_{\alpha/2}(n\hat{\rho}_n, n - n\hat{\rho}_n + 1), \mathsf{B}_{1-\alpha/2}(n\hat{\rho}_n + 1, n - n\hat{\rho}_n)]$$

*where $\mathsf{B}_\alpha(a, b)$ denotes the $\alpha^{th}$ quantile of a Beta distribution with parameters $a, b$. Then we have that:*

$$\Pr\left(\rho(\boldsymbol{x}) \in \hat{C}_\alpha(n, \hat{\rho}_n)\right) \geq 1 - \alpha.$$

*We can control the width of the confidence interval to $L \in (0,1)$ by estimating responsiveness with $n \geq N^{\min}(\alpha, L)$ reachable points where*

$$N^{\min}(\alpha, L) := \min\left\{n \in \mathbb{N} : \max_{s \in [n]}\left|\hat{C}_\alpha\left(n, \frac{s}{n}\right)\right| \leq L.\right\}$$

---

**Example 4** (Estimating Responsiveness for Feature Attribution).    *Consider a task where we need to identify salient features for a prediction in a lending task where $\mathcal{Y} = \{\mathsf{approve}, \mathsf{deny}\}$ [9]. If we were to do this based on responsiveness with a 0.05 margin of error where $f(\boldsymbol{x}) = \mathsf{deny}$, for each feature $j \in [d]$, we would set the parameters as follows:*

- $\hat{Y}_{\boldsymbol{x}}^{\text{reach}} = \{\mathsf{approve}\}$
- $A(\boldsymbol{x})$: *only allow feature $j$ and features linked via downstream effects to change*
- $\alpha = 0.05$, $L = 0.1$, *which implies $N^{\min}(\alpha, L) = 402$*

*and estimate responsiveness of the prediction with respect to interventions on each feature, identifying the most responsive features to report in mandated explanations (i.e., adverse action notice in the U.S.).*

---

In certain cases, we may wish to test if the responsiveness of predictions exceeds a threshold value $\varepsilon \in (0, 1)$. We may want to either identify points with extremely low responsiveness (e.g, detecting preclusion) or with high responsiveness (e.g., detecting gaming).

---

**Proposition 5** (Testing).    *Consider testing whether the responsiveness of a prediction for a model, $f$, at a point $\boldsymbol{x}$ exceeds a threshold value $\varepsilon > 0$ using the following hypotheses:*

$$H_0 : \; \rho(\boldsymbol{x}) \geq \varepsilon \Leftrightarrow \text{at least } 100 \cdot \varepsilon\% \text{ of interventions lead to target prediction } f(\boldsymbol{x}) \in \hat{Y}_{\boldsymbol{x}}^{\text{reach}}$$
$$H_1 : \; \rho(\boldsymbol{x}) < \varepsilon \Leftrightarrow \text{less than } 100 \cdot \varepsilon\% \text{ of interventions lead to target prediction } f(\boldsymbol{x}) \in \hat{Y}_{\boldsymbol{x}}^{\text{reach}},$$

*Given a sample of $n$ reachable points $\hat{X}_n$, let $\hat{\rho}_n$ denote the responsiveness estimate and $\rho_{2\alpha}^U(n, \hat{\rho}_n) := \mathsf{B}_{1-\alpha}(n\hat{\rho}_n + 1, n - n\hat{\rho}_n)$ denote the upper bound of the confidence interval $\hat{C}_{2\alpha}(n, \hat{\rho}_n)$, where $\alpha \in (0, 1)$ is the confidence parameter. In this case, we claim that the responsiveness is less than $\varepsilon$ whenever*

$$\rho_{2\alpha}^U(n, \hat{\rho}_n) < \varepsilon \iff \text{Reject } H_0$$

*Then, the probability of an* incorrect unresponsiveness claim *is bounded by* the confidence level $\alpha$:

$$\Pr(\rho_{2\alpha}^U(n, \hat{\rho}_n) < \varepsilon \mid \rho(\boldsymbol{x}) \geq \varepsilon) \leq \alpha.$$

*We calculate the minimum sample size, $N^{\min}$, that allows us to* correctly claim unresponsiveness *with probability $1 - \beta$ when the difference between $\varepsilon$ and $\rho$, true responsiveness, is at least $\Delta \in (0, \varepsilon)$:*

$$N^{\min}(\alpha, \beta, \varepsilon, \Delta) := \min\left\{n \in \mathbb{N} : \mathsf{F}(\mathsf{B}_\alpha(n\varepsilon, n - n\varepsilon); n(\varepsilon - \Delta), n - n(\varepsilon - \Delta) \geq 1 - \beta\right\}$$

*Here, $\mathsf{F}(\cdot; a, b)$ is the cumulative beta distribution function with parameters $a$ and $b$.*

---

**Example 6** (Testing Robustness to Random Fluctuations). *Consider testing if the predictions of a sepsis prediction model, $f$, developed by a third party are stable w.r.t. natural variations in clinical measurements using the medical devices of the local hospital. For each non-septic patient with features $\boldsymbol{x}$, we wish to limit the false alarms due to insignificant variation in certain measurements to at most 10%. We could set the parameters as follows:*

- $\hat{Y}_{\boldsymbol{x}}^{\mathrm{reach}} = \{\texttt{sepsis}\}$
- $A(\boldsymbol{x})$ *is such that* $a_{\texttt{systolic\_bp}} \in [-5\ mmHg, +5\ mmHg]$, *and*
  $a_{\texttt{bilirubin}} \in [-0.1 x_{\texttt{bilirubin}}\ mg/dL, 0.1 x_{\texttt{bilirubin}}\ mg/dL]$
- $\varepsilon = 0.1$, $\alpha = 0.01$, $\beta = 0.2$, $\Delta = 0.05$, *which would imply* $N^{\min}(\alpha, \beta, \varepsilon, \Delta) = 254$.

*Suppose that we observe 4 sepsis predictions in a set of $n = 254$ reachable points $\hat{X}_n$. Then, we have $\rho_{2\alpha}^{U} \approx 0.045 < \varepsilon$, thus we claim that the model is robust – allowing up to 10% predictions sensitive to random fluctuations – with probability of the false robustness claim $\alpha = 1\%$, and the probability of a correct robustness claim $1 - \beta = 80\%$ when the true responsiveness is at most $\varepsilon - \Delta = 0.05$.*

Propositions 3 and 5 draw on the fact that $\hat{S}_n \sim \mathsf{Bin}(n, \rho(\boldsymbol{x}))$ given an i.i.d. sample. Thus, we can construct confidence intervals on $\rho(\boldsymbol{x})$ using the exact method [11] and numerically compute $N^{\min}$. Although these results provide a guideline for how one might choose the sample size $n$, there is a strict lower bound on $n$ to avoid a trivial testing procedure that fails to reject $H_0$ for all $\boldsymbol{x}$:

**Remark 7.** *Given the $H_0$ and $H_1$ in Proposition 5 with confidence parameter $\alpha \in (0, 1)$,*

$$\text{Reject } H_0 \implies n > \log \alpha / \log(1-\varepsilon)$$

In other words, $n > \log \alpha / \log(1-\varepsilon)$ is a necessary condition to identify an unresponsive prediction. If this condition is not satisfied, we are not able to reject $H_0$, even if $\hat{\rho}(\boldsymbol{x}) = 0$.

# 4 Use Cases for Responsiveness Testing

We will demonstrate how our machinery can promote safety in model development or model auditing by estimating the responsiveness of predictions. We choose use cases in salient applications where we can build models with real-world datasets and highlight failure modes of responsiveness. We include additional details in Appendix C.

**Detecting Fixed Predictions in Recidivism Prediction Tools**  Many recidivism prediction models are designed to use features that cannot readily change – e.g., age and sex [see e.g., 4, 15, 26, 35], which assign more accurate risk predictions. These models tend to predict that defendants with certain characteristics beyond their control will recidivate *by default* – i.e., regardless of their charges or criminal history. As an example, we point to a risk score developed by the Pennsylvania Sentencing Commission [48] which assigns fixed predictions to male defendants under 21. This oversight perpetuates disproportionate harm against a vulnerable population, and was included in a model that took over five years to be developed by a panel of statisticians (with regular public feedback opportunities) before being implemented. Here, we show that our machinery could have revealed this via a simple audit in less than ten minutes.

We work with a sample of prisoners from New York compiled by the U.S. Department of Justice [56], which contains $n = 29{,}400$ and $d = 20$ features related to their age, sex, and criminal history (note that we do not include race). Here, the label is $y_i = 1$ if a defendant $i$ is rearrested within three years of release. We follow common practice [14, 61] and apply a standard 80-20 train-test split to fit and evaluate a logistic regression model (train/test AUC of 0.704/0.702). We test that this model assigns fixed predictions with respect to hypothetical interventions that "clear" criminal history – i.e., so that each defendant predicted to recidivate would be able to overturn their prediction by clearing features related to criminal history. We consider a test where $\varepsilon = 0.1$, $\alpha = 0.05$, $\beta = 0.2$, and target a resulting $\mathcal{Y}_i = 1$. We say that a prediction is "fixed" when $\Pr(\rho(\boldsymbol{x})) < 0.01$. Our intervention sets contains of 30 constraints – which capture changes to criminal history and their downstream effects (e.g., setting $\texttt{n\_prior\_arrests} = 5 \to 0$ would set $\texttt{prior\_arrests\_for\_felony} = 1 \to 0$). We construct reachable sets with 20 samples per point, satisfying Remark 7.

In Fig. 2, we show the distribution of fixed points. The model predicts that 18,614 individuals will recidivate on the training test of which 15,986 are assigned fixed predictions. We can also see that

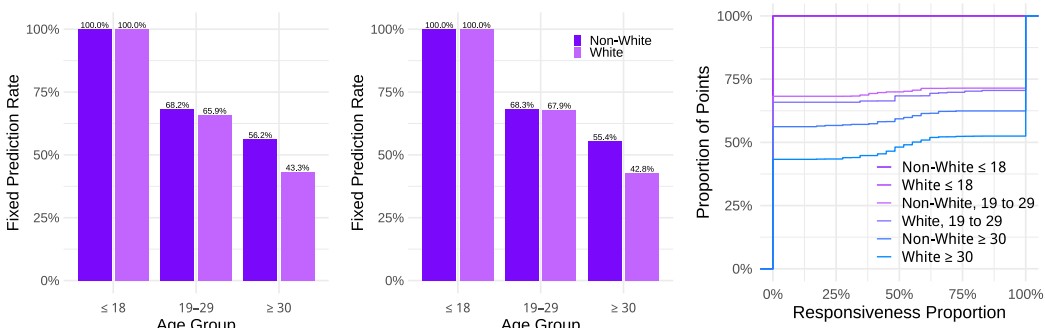

**Figure 2:** Distribution of unresponsive predictions in demographic groups. Left: Train sample. Middle: Test sample. Right: CDF of responsiveness proportion by demographic group

it follows patterns of prior recidivism models such as using age as a crucial indicator, and having a disproportionate impact across racial groups. We see that $100\%$ of all prisoners under the age of 18 are assigned fixed predictions. This is consistent with prior work showing that lower age is more correlated with a higher likelihood of models predicting recidivism [61]. We also see that non-white prisoners are assigned fixed predictions at a higher rate than white prisoners, especially in the $\geq 30$ age group. We see further evidence that age and ethnicity govern recidivism in the left-most plot. This provides further details on the relationship between age and race: as the age increases, race becomes a more important factor in determining if that prisoner will have recourse. Our methodology has (1) detected multiple failure modes of the model – racial bias and assigning fixed predictions, specifically disproportionately assigning fixed predictions across ethnic groups, and, importantly, (2) enabled finding these failures during model development.

**Testing Counterfactual Invariance in Organ Transplant Prioritization**   Predictive statistical models are routinely used in allocation of organ transplants [23]. Recently, they have attracted scrutiny both from the public and the academic circles because of their potential to assign fixed predictions, e.g., with evidence of lower access to transplants for younger patients [2, 43], and simulation studies showing that cancer patients are less likely to receive high prioritization [3].

We consider Transplant Benefit Score (TBS), a system used to prioritize transplants in the UK since 2018. We aim to test a basic monotonicity condition [5, 24] that the model should assign higher prioritization scores to a counterfactual patient with cancer, compared to the initial score of the patient without cancer. According to domain experts [3], having all other features fixed, getting cancer should increase the priority. Testing this system is challenging, as it comprises several submodules: two Cox proportional hazard regression model to predict *need* – survival without transplant – and two models to predict *utility* – survival with the transplant over the course of five years. We have the following component survival functions:

$$f_{\text{need}}^{\text{c}}(\boldsymbol{x}) = \sum_{t=1}^{T} S_{0,\text{need}}^{\text{c}}(t)^{\exp(\boldsymbol{\beta}_{\text{need}}^{\text{c}\top}(\boldsymbol{x}-\boldsymbol{\mu}_{\text{need}}^{\text{c}}))}, \qquad f_{\text{need}}^{\text{nc}}(\boldsymbol{x}) = \sum_{t=1}^{T} S_{0,\text{need}}^{\text{nc}}(t)^{\exp(\boldsymbol{\beta}_{\text{need}}^{\text{nc}\top}(\boldsymbol{x}-\boldsymbol{\mu}_{\text{need}}^{\text{nc}}))}$$

$$f_{\text{utility}}^{\text{c}}(\boldsymbol{x}) = \sum_{t=1}^{T} S_{0,\text{utility}}^{\text{c}}(t)^{\exp(\boldsymbol{\beta}_{\text{utility}}^{\text{c}\top}(\boldsymbol{x}-\boldsymbol{\mu}_{\text{utility}}^{\text{c}}))}, \quad f_{\text{utility}}^{\text{nc}}(\boldsymbol{x}) = \sum_{t=1}^{T} S_{0,\text{utility}}^{\text{nc}}(t)^{\exp(\boldsymbol{\beta}_{\text{utility}}^{\text{nc}\top}(\boldsymbol{x}-\boldsymbol{\mu}_{\text{utility}}^{\text{nc}}))}$$

where c and nc indicate models applied to patients with cancer and without, respectively, $S_{0,\cdot} : \mathbb{N} \rightarrow [0,1]$ for $t \in [T]$ for some $T \in \mathbb{N}$ are pre-defined baseline hazard functions, and the vectors $\beta$ and $\mu$ are the corresponding model parameters and data normalizers, respectively. The final TBS score is computed as $f_{\text{TBS}}(\boldsymbol{x}) = f_{\text{utility}}^{\boldsymbol{x}_{\text{cancer}}}(\boldsymbol{x}) - f_{\text{need}}^{\boldsymbol{x}_{\text{cancer}}}(\boldsymbol{x})$. An inspection of model coefficients $\boldsymbol{\beta}$ does not yield a simple answer on whether the system preserves monotonicity, especially as getting cancer involves a modification of several features at once, and using different models, $\boldsymbol{\beta}^{\text{c}}$.

To verify violations of monotonicity, we generate a cohort of 1,000 patients without cancer using a probabilistic model designed to mimic a prior simulation generated based on real patients [3]. We provide details on the dataset generation in Appendix C.3. We define two intervention sets: *"small"* in which we assign a patient to have a cancer with at most 2cm tumor size, and *"large"* with at

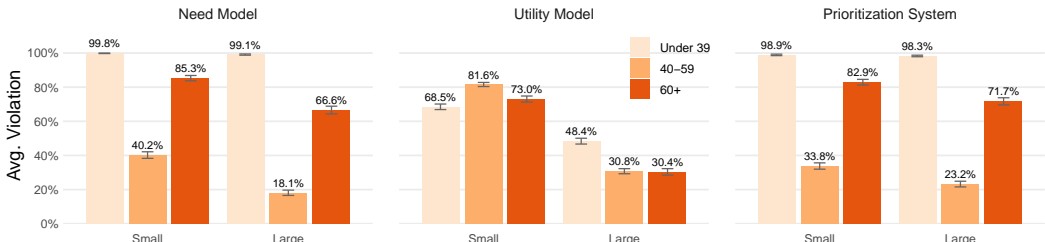

**Figure 3:** Average proportion of predictions that violate monotonicity across a cohort of simulated non-cancer patients, across two intervention sets in which counterfactual simulated patients are assigned cancer with size of either < 2cm ("small") or < 5cm ("large"). Error bars show 95% confidence interval around average violation across the simulated cohort.

| Procedure | Description | Model Pool | | % Resp. (Perceived) | | | % Resp. (True) | | | AUC | | |
| | | # Models | # Cert. Robust | Train | Test | Valid | Train | Test | Valid | Train | Test | Valid |
|---|---|---|---|---|---|---|---|---|---|---|---|---|
| Manual | Train Models with Immutable Features | 370 | 370 | 0.0% | 0.0% | 0.0% | 0.0% | 0.0% | 0.0% | 0.531 | 0.570 | 0.581 |
| Convex | Consider Responsiveness w.r.t Convex Perturbation Check | 901 | 687 | 0.3% | 0.0% | 0.9% | 56.2% | 57.1% | 55.9% | 0.743 | 0.754 | 0.759 |
| Exact | Evaluate Responsiveness w.r.t Exact Actions | 901 | 76 | 9.6% | 9.9% | 9.3% | 9.6% | 9.9% | 9.3% | 0.722 | 0.727 | 0.734 |

**Table 2:** We report results for the model with the highest validation AUC among *Considered* models: $\leq 10\%$ "Bot" predictions with certified responsiveness $\geq \varepsilon = 0.05$. *% Responsive* show % of "Bot" predictions with responsiveness $\geq \varepsilon = 0.05$ under the procedure's reachable set (*Perceived*) and the exact reachable set (*True*). We see that Convex under-reports model responsiveness and selects models prone to gaming.

most 5cm tumor size. Each intervention involves changing the disease indicator `primary_disease`, and the `max_tumor_size`, `tumor_number` features. Moreover, we use a random-effect response $r(x; a)$ which simulates natural variation in liver parameters such as the albumin level (see Appendix C.3). We measure and report average responsiveness $\hat{\rho}(x)$, where the prediction set of interest $\hat{Y}_x^{\text{reach}} = \{y \mid y < f(x)\}$ is those predictions which violate the monotonicity condition. Thus, in this case, responsiveness represents the proportion of *violations*.

We summarize our results in Fig. 3, shown separately for each submodule. These results show that (1) even inspecting individual components does not paint the full picture of model safety. Indeed, the *need* model (left) shows low average violation for the middle age group, but the *utility* (middle) shows significantly higher levels of violation. As the final score is a combination of both, it is unclear which result will be more important. The final TBS score (right), in the end, shows low violation in the middle age group. We can also see (2) that our procedure in a simulated cohort reveals that both younger and older patients could have their TBS scores *decreased* after getting cancer. Our tools flag this concrete safety issue on aggregate at the system level, enabling model developers to test responsiveness individually for each patient, and generate test cases for iterative model development.

**Preventing Gaming in Content Moderation**   Modern approaches for content moderation rely on machine learning models to limit misinformation or harassment at scale [20, 22]. In such settings, we often build models to predict if user account belong to a "bot" or "human", using these predictions to guide follow-up actions (e.g., human review or verification) to facilitate a more pleasant online experience [38]. Bot detection is difficult *at scale* – since many accounts lack substantial data, models must assign millions of predictions from a limited number of features that is available among all users. At the same time, we want to deploy models that are robust to manipulation – so that "bots" cannot skirt detection by "gaming" their account history or characteristics. The primary difficulty arises from the lack of available features, necessitating models to utilize a majority of them, thereby reducing robustness. In essence, the problem of building a robust model is akin to building a model that is unresponsive with respect to a realistic attack model.

We consider a task to detect bots derived from the `twitterbot` dataset [19] with $n = 3{,}431$ accounts and $d = 22$ features on their account characteristics (e.g., `n_tweets`, `inactivity`, `tweets_from_mobile_device`). To build realistic attack models, we capture feasible interventions through 4 non-separable constraints like enforcing `n_tweets = 0` when `inactivity = 1` (and

vice versa), and only allowing changes in `tweets_from_mobile_device` when `n_tweets` increases as one must upload a new post from their mobile device. We also assume some features such as `num_followers_leq_1000` are not actionable. Given the intervention model, we use an approach inspired by Zhang et al. [62] in which we construct classifiers that are robust to manipulation by penalizing or excluding certain features. We train a pool of penalized logistic regression models over a large grid of $l_1$ and $l_2$ parameters using `glmnet` [17]. We train models and assess their robustness to gaming through three approaches:

- Manual: We only use immutable features. This represents a baseline approach, which ensures robustness, but should attain low utility due to not utilizing all the available information.
- Convex: We use all features, but consider the convex relaxation of the intervention model to measure responsiveness, which is a common approach in robustness [see, e.g., 51, for a discussion].
- Exact: We use all features and consider the exact intervention model.

In Table 2, we report the results of the model that achieves the highest validation AUC among robust models – less than 10% of gameable predictions – that we train. Overall, our results highlight practical challenges when building a well-performing model that also limits gaming. Although models trained under the Manual procedure were all robust, they performed poorly with a Test AUC of 0.570. We also see that verifying responsiveness using a convex relaxation of the reachable set returns a well-performing model that *appears* robust. In fact, the test AUC of the model under the Convex regime (0.754) exceeds that of the model chosen under Exact (0.727). However, we see that the Convex procedure severely under-reports responsiveness: the perceived proportion of responsive points is near 0 in all three splits of the data, but, when verified against the actual reachable set, we see that the proportion of responsive points surges to $> 50\%$.

These results (1) show that there may exist a well-performing model that is robustness to gaming without additional adversarial training and (2) highlights the importance of validating responsiveness with respect to accurate interventions. Procedures like Convex can lead to unaccounted harm, where a model that is thought as robust can be deployed, only to be vulnerable to gaming.

## 5 Concluding Remarks

Over the past century, we have developed numerous practices to create and deploy technology that impacts people [44]—from tests that can be automated to standards that can be enforced. Even as machine learning models are routinely used to automate predictions that affect people, our practices are still in their infancy. Our work offers a concrete starting point to apply these approaches under imperfect data and action-induced distribution shift by presenting practitioners with machinery that can reliably detect failures in prediction responsiveness.

One of the benefits of our machinery is that it pairs each failed test with a subset of reachable points, which can support downstream tasks such as debugging, regression testing, or improving the specification of constraints and distributions of interventions. These points are also useful as counterexamples in tasks where we wish to falsify a claim (e.g., "the model will not assign a prediction that could be gamed"), including interactive settings where individuals adapt strategically. Our machinery can output such points, but is not designed to do so efficiently, as the points are uniformly distributed. In such cases, an importance sampling approach that accounts for the decision boundary may be more appropriate [46].

**Limitations** Our framework relies on practitioners specifying intervention constraints and downstream effects based on domain knowledge, documentation, or policy rules. While this enables broad applicability, it does not account for cases where causal relationships must be learned from data. Our method also does not infer constraints or causal structure automatically. Additionally, the sampling procedure may be computationally intensive in high-dimensional or tightly constrained settings, though this cost is amortized by reuse across models. Finally, our uniform sampling approach prioritizes coverage over efficiency; future work could explore adaptive or importance-based strategies for more efficient test generation.

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

# Reliable Models via Responsiveness Verification

## Supplementary Materials

# A    Supplementary Material for Section 1

**Related Work**    Our work is motivated by practical challenges in responsiveness that have broadly motivated work in adversarial robustness [21, 39], strategic classification [12, 18, 25, 37, 42], and counterfactual invariance [34, 53, 58]. Our machinery aims to detect these issues rather than resolve them in model development [c.f. work in strategic classification and robustness, e.g., 12, 16, 18, 25, 30, 37, 42]. To this end, we test with the same kinds of measures used in validation literature [52, 59]. Our work underscores how we can reap benefits from measuring responsiveness of models with semantically meaningful features – e.g., model selection [8] or identifying examples for debugging [55]. Our machinery provides a general way to enforce a rich set of semantic constraints for any model class.

Our work builds on a growing body of research on the reliability of individual predictions [see e.g., 32, 40, 41, 45]. Our work is closely related to a recent stream of work on *recourse verification* – i.e., a formal validation procedure to test if a model *can* provide recourse to its decision subjects [see e.g., 9, 33, 36]. Our approach builds on an idea introduced in Kothari et al. [33], who present a method to enumerate reachable points box to certify preclusion – i.e., that a model assigns predictions that cannot change. Their methods can output a deterministic guarantee of responsiveness but is restricted to datasets with discrete features and deterministic actions. Our methods to estimate responsiveness overcome these limitations by sampling a set of reachable points. This approach applies to tasks with discrete or continuous features, and can return estimates that support a broader class of model validation tasks.

# B  Supplementary Material for Section 3

## B.1  Uniform Sampling of Reachable Points

The main technical challenge in testing respon-
siveness is that it relies on a uniform sample
of interventions $\boldsymbol{a} \sim \mathsf{Uniform}[A(\boldsymbol{x})]$, which is
challenging due to the unconventional structure
of the intervention set – i.e., some features are
discrete while others could be continuous, with
certain combinations being infeasible.

We present a sampling procedure to yield a
set of reachable points in Algorithm 1. In
Line 3, given a subset of features $S$, we
first sample an intervention $a_j$ for each ac-
tionable feature $j \in [d] \setminus D$ by calling the
$\mathsf{SampleInterv}(\boldsymbol{x}, j, C)$ routine. After sampling
all interventions, we check if the resulting $\boldsymbol{a}$ is feasible under constraints in the block (Line 4) by
solving a discrete optimization problem: $\min_{\boldsymbol{a}' \in A(\boldsymbol{x})} \mathbb{I}[\boldsymbol{a}' = \boldsymbol{a}]$ s.t. $\boldsymbol{a}'$ satisfies $C$. We formulate
$\mathsf{CheckFeasibility}(\boldsymbol{x}, \boldsymbol{a}, C)$ using a mixed integer program and include a formulation in Appendix B.
In Line 5, we then sample values for each downstream features by calling the $\mathsf{SampleEffect}(\boldsymbol{a}, k, C)$
routine and add $\boldsymbol{x} + \boldsymbol{a} + \boldsymbol{r}$ to $\hat{X}_i$, the reachable set (Line 6).

---

**Algorithm 1** Sampler for Reachable Sets

**Require:** $\boldsymbol{x} \in \mathcal{X}$                      *Point*
**Require:** $n \geq 1$            *Sample Size*
**Require:** $C$                 *Constraints*
**Require:** $D \subseteq [d]$     *Downstream Features*
     $\hat{X} \leftarrow \varnothing$
1: **repeat**
2:    $\boldsymbol{a} \leftarrow \boldsymbol{0}$
3:    $a_j \leftarrow \mathsf{SampleInterv}(\boldsymbol{x}, j, C)$ for $j \in [d] \setminus D$
4:    **if** $\mathsf{CheckFeasibility}(\boldsymbol{x}, \boldsymbol{a}, C_S)$ **then**
5:      $r_k \leftarrow \mathsf{SampleEffect}(\boldsymbol{a}, k, C)$ for $k \in D$
6:      $\hat{X} \leftarrow \hat{X} \cup \{\boldsymbol{x} + \boldsymbol{a} + \boldsymbol{r}\}$
7:    **end if**
8: **until** $|\hat{X}| = n$
**Output:** $\hat{X}$

---

We improve the efficiency of the sampling procedure by proposing candidates that obey feature-
level constraints (integrality, monotonicity, bounds) in the $\mathsf{SampInterv}$ routine – e.g., if feature $j$ is
integer-valued, bounded to $B$, and monotonically increasing, we sample from $\mathsf{Uniform}(x_j, B)$. After
sampling based on feature-level constraints, we use $\mathsf{CheckFeasibility}(\boldsymbol{x}, \boldsymbol{a}, C_S)$ to ensure that they
obey joint actionability constraints like encoding constraints. Given $\boldsymbol{a}$, we then sample downstream
effects. For deterministic effects, we compute the appropriate feature response value $\boldsymbol{r}(\boldsymbol{a}; \boldsymbol{x})$ directly.
For stochastic effects, we sample based from the specified condition distribution $\boldsymbol{r}(\boldsymbol{a}; \boldsymbol{x}) \sim \mathbb{P}_{\boldsymbol{x}}(\boldsymbol{a})$.
By default, we sample from a uniform distribution of feasible values.

We also execute Algorithm 1 over subsets of features that are independent with respect to interventions
and downstream effects. We determine these subsets programmatically by identifying if a pair features
$j \neq j' \in [d]$ are coupled through constraints or distributions (e.g., if $a_j$ and $a'_j$ are linked directly or
indirectly – through another feature $a_k$). Given a graph that encodes this information for all $j, j' \in [d]$,
we can construct a *maximally independent* partition of features – i.e., a set of $k \leq d$ feature subsets
$\mathcal{M} := \{S_1, \ldots, S_k\}$ such that $A(\boldsymbol{x}) = \prod_{S \in \mathcal{M}} A_S(\boldsymbol{x}_S)$, where $A_S$ specifies intervention constraints
that apply to $\boldsymbol{x}_S$. Partitioning allows us to independently sample interventions within each subset,
which considerably improves sampling efficiency.

## B.2  Description of Routines in Algorithm 1

Here we provide further details on each of the routines referenced in Algorithm 1.

### Description of the $\mathsf{SampleInterv}$ Routine

The $\mathsf{SampleInterv}$ routine is designed to sample feasible values across features. Given a point $\boldsymbol{x}$, a
feature $j \in [d]$ and a set of constraints as defined by the intervention model $C$, $\mathsf{SampleInterv}(\boldsymbol{x}, j, C)$
samples an intervention $a_j \sim \mathsf{Unif}\{a'_j \mid \boldsymbol{a}' \in A(\boldsymbol{x})\}$. The procedure is designed to sample as
efficiently as possible in this setting by enforcing all constraints at the feature level: integrality,
monotonicity, bounds on the value of $x_j$, and bounds on the value of $a_j$. If feature $j$ is discrete, we
take a uniform sample from

$$[\mathrm{LB}_j(\boldsymbol{x}), \mathrm{UB}_j(\boldsymbol{x})]_{\mathbb{Z}} = [\mathrm{UB}_j(\boldsymbol{x})] \setminus [\mathrm{LB}_j(\boldsymbol{x})].$$

If feature $j$ is continuous, we take a uniform sample from

$$[\mathrm{LB}_j(\boldsymbol{x}), \mathrm{LB}_j(\boldsymbol{x})].$$

We define the lower and upper bounds for the intervention on $j$, $\mathrm{LB}_j(\boldsymbol{x})$ and $\mathrm{UB}_j(\boldsymbol{x})$ as:

$$\mathrm{UB}_j(\boldsymbol{x}) = \mathbb{I}[j \uparrow] \cdot (ub_j - x_j)$$

$$\mathrm{LB}_j(\boldsymbol{x}) = \mathbb{I}[j \downarrow] \cdot (x_j - lb_j)$$

Here, $\mathbb{I}[j \uparrow] = 1$ if $j$ can increase, $\mathbb{I}[j \downarrow]$ if $j$ can decrease and $lb_j, ub_j$ are bounds on feature $j$ (note that $x_j \in [lb_j, ub_j]$).

### Description of CheckFeasibility **Routine**

CheckFeasibility determines whether $\boldsymbol{a}'$, the sampled intervention, is feasible under the constraint set $C$. Although SampleInterv ensures that each $a_j$ for $j \in [d]$ abides by feature level constraints like integrality, monotonicity and bounds, we must additionally ensure that $\boldsymbol{a}'$ does not violate non-separable constraints.

More formally, given $\boldsymbol{x}$, a sampled intervention $\boldsymbol{a}'$ and a set of constraints $C$, CheckFeasibility solves the following problem:

$$\min_{\boldsymbol{a} \in A(\boldsymbol{x})} \quad \mathbb{I}[\boldsymbol{a}' = \boldsymbol{a}] \quad \text{s.t. } \boldsymbol{a} \text{ abides by } C \tag{1}$$

We implement Eq. (1) as a mixed-integer program that consists of a baseline formulation – enforcing separable constraints like bounds and monotonicity – and additional constraints, which enforce non-separable constraints, and optionally, downstream effects. The baseline formulation has the form:

$$\min \sum_{j \in [d]} (a_j^+ + a_j^-) \tag{2a}$$

$$\text{s.t.} \quad a_j = a_j' \qquad j \in [d] \quad \textit{intervene with } \boldsymbol{a}' \tag{2b}$$

$$a_j^+, a_j^- \in \mathbb{R}_+ \qquad j \in [d] \quad \textit{positive, negative compoenets of } a_j \tag{2c}$$

$$a_j = a_j^+ - a_j^- \qquad j \in [d] \quad \textit{absolute value reconstruction} \tag{2d}$$

$$\sigma_j \in \{0, 1\} \qquad j \in [d] \quad \textit{sign of } a_j \tag{2e}$$

$$a_j^+ \geq a_j \qquad j \in [d] \quad \textit{positive component of } a_j \tag{2f}$$

$$a_j^- \geq -a_j \qquad j \in [d] \quad \textit{negative component of } a_j \tag{2g}$$

$$a_j^+ \leq \mathrm{UB}_j(\boldsymbol{x})\sigma_j \qquad j \in [d] \quad \textit{only 1 of } a_j^+ \textit{ or } a_j^- \textit{ can be positive} \tag{2h}$$

$$a_j^- \leq \mathrm{LB}_j(\boldsymbol{x})(1 - \sigma_j) \quad j \in [d] \quad \textit{only 1 of } a_j^+ \textit{ or } a_j^- \textit{ can be positive} \tag{2i}$$

$$\boldsymbol{a} \in A(\boldsymbol{x}) \qquad \textit{joint actionability constraints} \tag{2j}$$

The baseline formulation in Eq. (2) minimizes the $l_1$ norm of $\boldsymbol{a}$, splitting $\boldsymbol{a}$ into positive and negative parts $a_j^+, a_j^- \geq 0$ (2d), of which only one is non-zero. This allows us to use this baseline formulation for both sampling and enumeration. Here, $\sigma_j := \mathbb{I}[a_j > 0]$ is a boolean variable which we set to 1 when $a_j$ is positive to ensure that signed components can have a positive value through (2e).

(2b) stipulates that we intervene with $\boldsymbol{a}'$ – i.e., find an intervention $\boldsymbol{a}$ such that satisfies the remaining constraints and is equal to $\boldsymbol{a}'$. The remaining constraints enforces separable (constraint (2h), (2i)) and non-separable actionability constraints (constraint (2j)).

Below we provide two examples of non-separable actionability constraints and their explicit formulation in Eq. (2). For additional examples of how we can explicitly encode constraints into Eq. (2), refer to [33].

**Encoding Directional Linkage Constraints**   We often encounter features where intervening on them has a direct (and sometimes deterministic) effect on other features. For example, in Table 8, joint constraint 4 stipulates that urls_count increases at most as the change in num_tweets. Here, the "source variable" – the source of the effect – is num_tweets and the "target variable" – the feature affected – is urls_count.

We capture this effect, called *Directional Linkage*, by adding additional constraints to Eq. (2). Given source feature $k \in [d]$, a non-empty set of target features $T \subseteq [d] \setminus \{k\}$ and a scale vector $\boldsymbol{s} \in \mathbb{R}^{|T|}$, which captures the scale of the effect for each $l \in T$, we add the following constraints:

$$b_l - s_l \cdot a_k = 0 \tag{3}$$

$$c_l - a_l - b_l = 0 \tag{4}$$

for each target feature $l \in T$, where $b_l$ indicates the change in feature $l$ as a result of intervention $a_k$, and $c_l$ represents the aggregate change in $l$.

We can also substitute the equality in Eq. (3) with inequalities. The aforementioned example with `num_tweets` and `urls_count` is a case where the relationship is an inequality ($\leq$) and $s = 1$.

**Encoding Thermometer Encoding Constraints**    Datasets often include features that are based on thresholds. These features are often encoded like unary codes, a number of ones followed by zeros. For example, in Table 8, `age_of_account_geq` has a thermometer encoding with thresholds at 180, 365, 730 and 1825 days. Hence there are five possible encoding values:

1. $[0, 0, 0, 0]$: account is less than 180 days old

2. $[1, 0, 0, 0]$: account is older than 180 days but less than 365 days old

3. $[1, 1, 0, 0]$: account is older than 365 days but less than 730 days old

4. $[1, 1, 1, 0]$: account is older than 730 days but less than 1825 days old

5. $[1, 1, 1, 1]$: account is more than 1825 days old

Given an ordered set of feasible values $V$, like above, we also define a reachability matrix $E \in \{0, 1\}^{|V| \times |V|}$, where the $(i, j)$-th entry of $E$ is 1 when we can reach from the $i$-th element of $V$ to its $j$th element and 0 otherwise. Note that there are three possibilities for $E$: an upper triangular matrix, a lower triangular matrix of ones, or an all-one matrix. For example, `age_of_account_geq`, we also have a monotonicity constraint – age can only increase. So given the set of viable values (in order), the reachability matrix $E$ is an upper triangle matrix of ones (i.e., can reach $[1, 0, 0, 0]$ from $[0, 0, 0, 0]$, but not ther other way around).

Then, we add the following constraints to Eq. (2):

$$\sum_{k \in [|V|]} u_k = 1 \tag{5}$$

$$a_j = \sum_{k \in [|V|]} e_{j,k}(v_{k,j} - x_j)u_k \tag{6}$$

where $u_k = 1$ when resulting feature vector after the proposed intervention $\boldsymbol{a}'$ corresponds to the $k$-th encoding in $V$, $v_k$, 0 otherwise. $e_{j,k}$ indicates whether $v_k$ is reachable (based on $E$). Eq. (5) ensures that $\boldsymbol{a}'$ has a valid encoding and Eq. (6) computes the required change (if feasible).

### Description of SampleEffect **Routine**

The implementation SampleEffect changes based on the nature and relationships for the downstream effects we wish to sample:

- For deterministic downstream effects, we do not sample but calculate the effect directly as there is only one feasible value. We have implemented a baseline sampler for non-deterministic downstream effects, which takes a uniform sample from possible feature values and runs CheckFeasibility on the resulting final intervention $\boldsymbol{a} + \boldsymbol{r}$.
- For random or causal effects, we sample $\boldsymbol{r}(\boldsymbol{a}; \boldsymbol{x})$ from the specified distribution or model $\mathbb{P}_{\boldsymbol{x}}(\boldsymbol{a})$. Note that the parameters of the distribution need not be the same for all points.

### Partitioning for Efficiency

We run Algorithm 1 separately over subsets of features, rather than jointly over all features in $[d]$. These subsets are disjoint and are independent with respect to interventions and downstream effects. More formally, we call the collection of these independent subsets a *partition* $\mathcal{M} := \{S_1, S_2, \ldots, S_k\}$ of $[d]$ such that given two parts $S_m, S_n$, there are no joint constraints or downstream effects between all pairs $(p, q) \in S_m \times S_n$ of features. Another way to think about feature partitions would be as connected components in a graph, where features are nodes and edges represent joint constraints and/or downstream effects (i.e., $\exists$ edge $(p, q) \iff$ there are joint actionability constraints between $p$ and $q$).

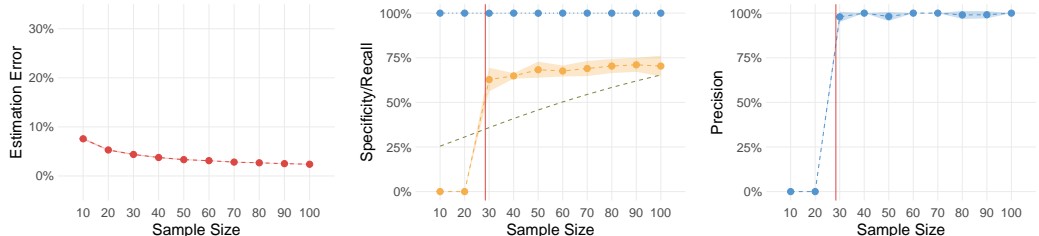

**Figure 4:** Convergence of responsiveness estimates and test metrics for a lending model built from the german dataset [13]. We compute the true responsiveness of all instances in the dataset by enumeration, build sampled reachable sets to estimate and test responsiveness ($\varepsilon = 0.1, \alpha = 0.05$). **Left**: *Absolute Estimation Error* ($|\hat{\rho}_n - \rho|$). **Middle**: *Specificity* ($P(\text{Claim Responsive} \mid \rho \geq \varepsilon)$, analogous to statistical power: $1 - \beta$) and *Recall* ($P(\text{Claim Unresponsive} \mid \rho < \varepsilon)$, analogous to confidence level $1 - \alpha$). The dotted line is the statistical power across different sample sizes $n$ given effect size $\Delta = 0.05$. **Right**: *Precision* ($P(\rho < \varepsilon \mid \text{Claim Unresponsive})$). Red lines in Middle and Right figures show the minimum sample size required to reject the null hypothesis given no positive observations: $\log \alpha / \log(1-\varepsilon)$ (Remark 7).

The benefit of sampling within partitions is two-fold:

- *Scalability*: We only execute CheckFeasibility when necessary (i.e., when the partition is larger than size 1. Moreover, we only discard infeasible samples within the partition, rather than throwing out the entire sampled intervention. This significantly decreases run time for sampling.
- *Implementation*: We can apply more efficient sampling procedures. In general, a dataset will have many kinds of features – e.g., continuous and discrete – with many different kinds of actionability constraints. However, subsets of features are likely to be similar. In effect, we can often find features that are not related to other features. Alternatively, we may find features that are all discrete and linked together by a single constraint (e.g., dummy variables with a one-hot encoding). Decomposition allows us apply different sampling procedures to each to sample more efficiently.

### B.3 Validation Study

**Convergence Guarantees** Our sampling-based procedure provides several statistical guarantees for our responsiveness estimate: $\hat{\rho}(\boldsymbol{x})$ is an unbiased estimator and the (absolute) estimation error tends to 0 as the sample size $n$ increases. For testing, our results in Proposition 5 state that the probability of *correctly identifying responsiveness* (Specificity) is at least $1 - \alpha$ and the probability of *correctly identifying unresponsiveness* (Recall) is at least $1 - \beta$ given $n \geq N^{\min}$. In practice, these guarantees imply that we can adapt tests to achieve any level of specificity or recall by setting the appropriate sample size.

We demonstrate these guarantees through an empirical study detailed in Appendix B. We work with a dataset with discrete features where we can enumerate all reachable points for each instance and compute ground-truth responsiveness. We use these to estimate the absolute estimation error ($|\hat{\rho}_n - \rho|$), specificity ($\Pr(\text{Claim Responsive} \mid \rho \geq \varepsilon)$) and recall ($\Pr(\text{Claim Unresponsive} \mid \rho < \varepsilon)$). In addition to verifying these guarantees, we investigate the precision ($\Pr(\rho < \varepsilon \mid \text{Claim Unresponsive})$) of our tests to gauge their reliability in action.

As shown in Fig. 4, the absolute estimation error decreases as $n$ increases and specificity remains above $1 - \alpha = 95\%$. We also observe the results in Remark 7, where both precision and recall are 0 for $n < \log \alpha / \log(1-\varepsilon)$ since the test fails to reject $H_0$ for all predictions (i.e., none flagged as unresponsive). For $n > \log \alpha / \log(1-\varepsilon)$, we see that the specificity of our test is above the statistical power (dotted line) computed at $n$. The precision of the test is also above 95% for $n > \log \alpha / \log(1-\varepsilon)$, indicating that our tests result in very few false positives (i.e., claiming unresponsiveness when the prediction is responsive).

These results reaffirm our statistical guarantees and highlight that we can achieve low estimation error and high test reliability with a relatively small sample size. For example, a sampled reachable set with $n = 30$ has 4.2% absolute estimation error and 97.9% precision on average across 5 trials, while taking up 85% less storage. As a result, even in cases where the intervention sets are discrete and can be enumerated, sampling can lead to a meaningful reduction in compute and storage instances.

In Fig. 4, we conduct a study on sample size $n$ to (1) validate our responsiveness estimation and testing procedure outlined in Section 3, and (2) determine their reliability under various sample sizes. We work with a discrete dataset, german, where we can fully enumerate reachable sets using the enumeration procedure from Kothari et al. [33]. The enumerated reachable sets provides ground truth responsiveness proportions. We compare our results to determine the error of our *estimation* procedure and the precision (in the main body, we refer to it as "reliability" for simplicity) of our *testing* procedure under two model classes: Logistic Regression (LR) and XGBoost (XGB).

The german dataset is a credit dataset originally compiled in 1994 that is publicly available through the UCI Machine Learning Repository [13]. It contains $n = 1,000$ de-identified instances, each representing a credit applicant. It includes $d = 20$ categorical or discrete features, providing insights into aspects such as loan history, demographic information (including gender, age, and marital status), occupation, and past payment behavior. The objective is to predict whether an applicant is a "good" ($y_i = 1$) or "bad" ($y_i = 0$) credit customer. We note that the dataset does not have missing values, and have adapted some feature names for clarity.

**Intervention Model**   We consider an intervention model where each applicant can intervene on current features like account balances, but not history nor credit related features. For example, Housing=Owner is not actionable since one cannot go from renting to buying without additional loans. This intervention model is conservative and is intended to capture indisputable actionability constraints. In total, our dataset contains 36 features of which 9 are actionable and 10 are mutable. There a total of four constraints: two Directional Linkage, and two Thermometer Encoding constraints:

- Directional Linkage constraints in this intervention model govern downstream effects on Age from 1) YearsAtResidence and 20 YearsEmployed≥1, which form a partition.
- Thermometer Encoding constraints enforce conceptual requirements in this dataset - 1) requiring CheckingAcct≥0=True to be reachable only if CheckingAcct_exists is also True, and 2) requiring SavingsAcct≥100=True to be reachable only if SavingsAcct_exists is True.

These lead to 31 partitions.

We present a list of all features and their corresponding feature-level constraints in Table 3 and list the non-separable joint constraints below it.

1. DirectionalLinkage: Actions on YearsAtResidence will induce actions on ['Age']. Each unit change in YearsAtResidence leads to a unit change in Age
2. DirectionalLinkage: Actions on YearsEmployed≥1 will induce actions on ['Age']. Each unit change in YearsEmployed≥1 leads to a unit change in Age
3. ThermometerEncoding: Actions on [CheckingAcctexists, CheckingAcct≥0] must preserve thermometer encoding of CheckingAcct., which can only increase. Actions can only turn on higher-level dummies that are off, where CheckingAcctexists is the lowest-level dummy and CheckingAcct≥0 is the highest-level-dummy.
4. ThermometerEncoding: Actions on [SavingsAcctexists, SavingsAcct≥100] must preserve thermometer encoding of SavingsAcct., which can only increase. Actions can only turn on higher-level dummies that are off, where SavingsAcctexists is the lowest-level dummy and SavingsAcct≥100 is the highest-level-dummy.

Lastly, we report model performance statistics for our LR and XGB model:

| Name | Type | LB | UB | Actionable | Sign | Joint Constraints | Partition ID |
|------|------|----|----|-----------|------|-------------------|--------------|
| Age | $\mathbb{Z}$ | 19 | 75 | No | | 1, 2 | 0 |
| YearsAtResidence | $\mathbb{Z}$ | 0 | 7 | Yes | + | 1 | 0 |
| YearsEmployed$\geq$1 | $\{0,1\}$ | 0 | 1 | Yes | + | 2 | 0 |
| CheckingAcct_exists | $\{0,1\}$ | 0 | 1 | Yes | + | 3 | 30 |
| CheckingAcct$\geq$0 | $\{0,1\}$ | 0 | 1 | Yes | + | 3 | 30 |
| SavingsAcct_exists | $\{0,1\}$ | 0 | 1 | Yes | + | 4 | 31 |
| SavingsAcct$\geq$100 | $\{0,1\}$ | 0 | 1 | Yes | + | 4 | 31 |
| Male | $\{0,1\}$ | 0 | 1 | No | | – | 1 |
| Single | $\{0,1\}$ | 0 | 1 | No | | – | 2 |
| ForeignWorker | $\{0,1\}$ | 0 | 1 | No | | – | 3 |
| LiablePersons | $\mathbb{Z}$ | 1 | 2 | No | | – | 4 |
| Housing=Renter | $\{0,1\}$ | 0 | 1 | No | | – | 5 |
| Housing=Owner | $\{0,1\}$ | 0 | 1 | No | | – | 6 |
| Housing=Free | $\{0,1\}$ | 0 | 1 | No | | – | 7 |
| Job=Unskilled | $\{0,1\}$ | 0 | 1 | No | | – | 8 |
| Job=Skilled | $\{0,1\}$ | 0 | 1 | No | | – | 9 |
| Job=Management | $\{0,1\}$ | 0 | 1 | No | | – | 10 |
| CreditAmt$\geq$1000K | $\{0,1\}$ | 0 | 1 | No | | – | 11 |
| CreditAmt$\geq$2000K | $\{0,1\}$ | 0 | 1 | No | | – | 12 |
| CreditAmt$\geq$5000K | $\{0,1\}$ | 0 | 1 | No | | – | 13 |
| CreditAmt$\geq$10000K | $\{0,1\}$ | 0 | 1 | No | | – | 14 |
| LoanDuration$\leq$6 | $\{0,1\}$ | 0 | 1 | No | | – | 15 |
| LoanDuration$\geq$12 | $\{0,1\}$ | 0 | 1 | No | | – | 16 |
| LoanDuration$\geq$24 | $\{0,1\}$ | 0 | 1 | No | | – | 17 |
| LoanDuration$\geq$36 | $\{0,1\}$ | 0 | 1 | No | | – | 18 |
| LoanRate | $\mathbb{Z}$ | 1 | 4 | No | | – | 19 |
| HasGuarantor | $\{0,1\}$ | 0 | 1 | Yes | + | – | 20 |
| LoanRequiredForBusiness | $\{0,1\}$ | 0 | 1 | No | | – | 21 |
| LoanRequiredForEducation | $\{0,1\}$ | 0 | 1 | No | | – | 22 |
| LoanRequiredForCar | $\{0,1\}$ | 0 | 1 | No | | – | 23 |
| LoanRequiredForHome | $\{0,1\}$ | 0 | 1 | No | | – | 24 |
| NoCreditHistory | $\{0,1\}$ | 0 | 1 | No | | – | 25 |
| HistoryOfLatePayments | $\{0,1\}$ | 0 | 1 | No | | – | 26 |
| HistoryOfDelinquency | $\{0,1\}$ | 0 | 1 | No | | – | 27 |
| HistoryOfBankInstallments | $\{0,1\}$ | 0 | 1 | Yes | + | – | 28 |
| HistoryOfStoreInstallments | $\{0,1\}$ | 0 | 1 | Yes | + | – | 29 |

**Table 3:** Intervention Model for the processed german dataset. **Type** indicates the feature type ($\mathbb{Z}$ for integer, $\{0,1\}$ for binary). **LB**, **UB** are the lower and upper bounds for the feature. **Actionable** indicates whether the feature is globally actionable. Non-actionable features are highlighted. **Sign** indicates monotonicity constraints – whether feature can only increase (+) or decrease (-). **Joint Constraints** are a list non-separable constraint indices (listed below table) it is tied to (if any). **Partition ID** indicates which partition the feature belongs to.

| | LR | | XGB | |
|---|---|---|---|---|
| | **Train** | **Test** | **Train** | **Test** |
| **AUC** | 0.807 | 0.768 | 0.819 | 0.7615 |
| **Expected Calibration Error** | 20.0% | 20.0% | 0.0% | 10.0% |
| **Error** | 27.2% | 28.0% | 21.9% | 23.0% |
| $n$ | 800 | 200 | 800 | 200 |
| $n_{\textbf{pos}}$ | 560 | 140 | 560 | 140 |
| $p$ | 70.0% | 70.0% | 70.0% | 70.0% |
| $n_{\textbf{clf\_pos}}$ | 738 | 186 | 615 | 120 |
| $n_{\textbf{clf\_neg}}$ | 62 | 14 | 185 | 80 |

**Table 4:** Additional model statistics of LR and XGB models for the german dataset

.

# C   Supplementary Material for Section 4

In this Appendix, we provide additional details and results for each of the use cases in Section 4.

## C.1   Detecting Fixed Predictions in Recidivism Prediction Tools

### C.1.1   Description of Dataset

We work with a large sample of defendants from New York state derived from the "Recidivism of Prisoners Released in 1994" dataset released by the U.S. Department of Justice [56], which contains $n = 29{,}400$ and $d = 20$ features about their criminal history. This dataset has been used in recidivism studies such as [40, 61]. Here, the label is $y_i = 1$ if a prisoner is rearrested within the 3 years of release from prison. We include 12 features explicitly related to criminal history, two immutable characteristics (`age` and `female`), and six mutable characteristics, four of which are actionable, do not provide additional information about criminal history.

- Criminal History Features: All features relating to `prior_arrests`, all features relating to `time_served`, `any_prior_prb_or_fine`
- Mutable: `edu_program_participication`, `voc_program_participation`, `drug_abuser`, `drug_treatment`, `alcohol_abuser`, `alcohol_treatment`

We bucketize `age_at_release` as follows:

- $\leq 16$
- 16 to 19
- 19 to 23
- 23 to 27
- 27 to 30
- 30 to 35
- 35 to 40
- 40 to 45
- $\geq 45$

### C.1.2   Intervention Model

**Intervention Model**   We consider an intervention model where each defendant can perform (1) actions that change actionable features about their participation in rehabilitation profile (e.g., participating in educational programs, setting `edu_program_participation` to `True`), and (2) hypothetical actions that would clear their criminal history (see below for detailed examples).

Our dataset contains 20 features of which 7 are actionable and 18 are mutable. The intervention model contains a total of 27 constraints: 24 Directional Linkage constraints, and three Reachability Constraints:

- *Criminal History Constraints.* Each of `prior_arrests=1`, `prior_arrests≥2`, and `prior_arrests≥5` has the same sets of constraints: Each `time_served` variable must decrease, `any_prior_prb_or_fine` must decrease, `prior_arrests_for_felony`, `prior_arrests_for_misdemeanor`, and `prior_arrests_for_general_violence` must decrease, and finally `no_prior_arrests` must be `True`. The associated ReachabilityConstraint forces `prior_arrests=1`, `prior_arrests≥2`, and `prior_arrests≥5` to only be able to reach `no_prior_arrests`, fully clearing arrest history and preventing the number of arrests from decreasing by 1.
- *Non-Criminal History Constraints:* Both `drug_abuser` and `alcohol_abuser` have a Reachability-Constraint with their corresponding `treatment` feature - this constraint ensures that `treatment` is only reachable if `abuser` is `True`.

Note that these create corresponding partitions (see Table 5): 0 (`alcohol` features), and 1 (`drug` features), 2 (`edu_program_participation`, which can only increase), 3 (`voc_program_participation`, which can only increase), 4 (`age_at_release`, immutable), 5 (`female`, immutable), and 6 (the criminal history constraints outlined above).

We present a list of all features and their corresponding feature-level constraints in Table 5.

| Name | Type | LB | UB | Actionable | Sign | Constraints | Partition ID |
|---|---|---|---|---|---|---|---|
| prior_arrests=1 | {0,1} | 0 | 1 | Yes | − | 2, 5, 8, 11, 14, 17, 20, 23, 25 | 6 |
| prior_arrests≥2 | {0,1} | 0 | 1 | Yes | − | 1, 4, 7, 10, 13, 16, 19, 22, 25 | 6 |
| prior_arrests≥5 | {0,1} | 0 | 1 | Yes | − | 3, 6, 9, 12, 15, 18, 21, 24, 25 | 6 |
| no_prior_arrests | {0,1} | 0 | 1 | No | | 25 | 6 |
| time_served≤1_year | {0,1} | 0 | 1 | No | | 1, 2, 3 | 6 |
| time_served_g_1_year | {0,1} | 0 | 1 | No | | 4, 5, 6 | 6 |
| time_served_g_2_years | {0,1} | 0 | 1 | No | | 7, 8, 9 | 6 |
| time_served_g_5_years | {0,1} | 0 | 1 | No | | 10, 11, 12 | 6 |
| prior_arrests_for_misdemeanor | {0,1} | 0 | 1 | No | | 13, 14, 15 | 6 |
| prior_arrests_for_felony | {0,1} | 0 | 1 | No | | 22, 23, 24 | 6 |
| prior_arrests_for_general_violence | {0,1} | 0 | 1 | No | | 16, 17, 18 | 6 |
| any_prior_prb_or_fine | {0,1} | 0 | 1 | No | | 19, 20, 21 | 6 |
| drug_abuser | {0,1} | 0 | 1 | No | | 26 | 0 |
| drug_treatment | {0,1} | 0 | 1 | Yes | + | 26 | 0 |
| alcohol_abuser | {0,1} | 0 | 1 | No | | 27 | 1 |
| alcohol_treatment | {0,1} | 0 | 1 | Yes | + | 27 | 1 |
| edu_program_participation | {0,1} | 0 | 1 | Yes | + | − | 2 |
| voc_program_participation | {0,1} | 0 | 1 | Yes | + | − | 3 |
| age_at_release | ℝ | 17.3 | 83.9 | No | | − | 4 |
| female | {0,1} | 0 | 1 | No | | − | 5 |

**Table 5:** Intervention model for the `rearrest_NY` dataset. **Type** indicates the feature type (ℝ for real numbers, {0, 1} for binary). **LB**, **UB** are the lower and upper bounds for the feature. **Actionable** indicates whether the feature is globally actionable. Non-actionable features are highlighted. **Sign** indicates monotonicity constraints – whether feature can only increase (+) or decrease (-). **Joint Constraints** are a list non-separable constraint indices (listed below table) it is tied to (if any). **Partition ID** indicates which partition the feature belongs to.

In this case, the intervention model must enforce a large set of deterministic downstream effects to maintain the semantic relationships between the features of the model while "clearing criminal history." In general, we would enforce these relationships through the sampling distribution. Given that they are deterministic effects, however, we enforce them by defining non-separable constraints. The final set of joint actionability constraints include:

1. DirectionalLinkage: Actions on priorarrests≥2 will induce actions on [timeserved≤1year]. Each unit change in priorarrests≥2 leads to a unit change in timeserved≤1year
2. DirectionalLinkage: Actions on priorarrests=1 will induce actions on timeserved≤1year. Each unit change in priorarrests=1 leads to a unit change in timeserved≤1year
3. DirectionalLinkage: Actions on priorarrests≥5 will induce actions on timeserved≤1year. Each unit change in priorarrests≥5 leads to a unit change in timeserved≤1year
4. DirectionalLinkage: Actions on priorarrests≥2 will induce actions on timeservedg1year. Each unit change in priorarrests≥2 leads to a unit change in timeservedg1year
5. DirectionalLinkage: Actions on priorarrests=1 will induce actions on timeservedg1year. Each unit change in priorarrests=1 leads to a unit change in timeservedg1year
6. DirectionalLinkage: Actions on priorarrests≥5 will induce actions on timeservedg1year. Each unit change in priorarrests≥5 leads to a unit change in timeservedg1year
7. DirectionalLinkage: Actions on priorarrests≥2 will induce actions on timeservedg2years. Each unit change in priorarrests≥2 leads to a unit change in timeservedg2years
8. DirectionalLinkage: Actions on priorarrests=1 will induce actions on timeservedg2years. Each unit change in priorarrests=1 leads to a unit change in timeservedg2years
9. DirectionalLinkage: Actions on priorarrests≥5 will induce actions on timeservedg2years. Each unit change in priorarrests≥5 leads to a unit change in timeservedg2years
10. DirectionalLinkage: Actions on priorarrests≥2 will induce actions on timeservedg5years. Each unit change in priorarrests≥2 leads to a unit change in timeservedg5years
11. DirectionalLinkage: Actions on priorarrests=1 will induce actions on timeservedg5years. Each unit change in priorarrests=1 leads to a unit change in timeservedg5years
12. DirectionalLinkage: Actions on priorarrests≥5 will induce actions on timeservedg5years. Each unit change in priorarrests≥5 leads to a unit change in timeservedg5years
13. DirectionalLinkage: Actions on priorarrests≥2 will induce actions on priorarrestsforfelony. Each unit change in priorarrests≥2 leads to a unit change in priorarrestsforfelony
14. DirectionalLinkage: Actions on priorarrests=1 will induce actions on priorarrestsforfelony. Each unit change in priorarrests=1 leads to a unit change in priorarrestsforfelony

15. DirectionalLinkage: Actions on `priorarrests≥5` will induce actions on `priorarrestsforfelony`. Each unit change in `priorarrests≥5` leads to a unit change in `priorarrestsforfelony`
16. DirectionalLinkage: Actions on `priorarrests≥2` will induce actions on `priorarrestsformisdemeanor`. Each unit change in `priorarrests≥2` leads to a unit change in `priorarrestsformisdemeanor`
17. DirectionalLinkage: Actions on `priorarrests=1` will induce actions on `priorarrestsformisdemeanor`. Each unit change in `priorarrests=1` leads to a unit change in `priorarrestsformisdemeanor`
18. DirectionalLinkage: Actions on `priorarrests≥5` will induce actions on `priorarrestsformisdemeanor`. Each unit change in `priorarrests≥5` leads to a unit change in `priorarrestsformisdemeanor`
19. DirectionalLinkage: Actions on `priorarrests≥2` will induce actions on `priorarrestsforgeneralviolence`. Each unit change in `priorarrests≥2` leads to a unit change in `priorarrestsforgeneralviolence`
20. DirectionalLinkage: Actions on `priorarrests=1` will induce actions on `priorarrestsforgeneralviolence`. Each unit change in `priorarrests=1` leads to a unit change in `priorarrestsforgeneralviolence`
21. DirectionalLinkage: Actions on `priorarrests≥5` will induce actions on `priorarrestsforgeneralviolence`. Each unit change in `priorarrests≥5` leads to a unit change in `priorarrestsforgeneralviolence`
22. DirectionalLinkage: Actions on `priorarrests≥2` will induce actions on `anypriorprborfine`. Each unit change in `priorarrests≥2` leads to a unit change in `anypriorprborfine`
23. DirectionalLinkage: Actions on `priorarrests=1` will induce actions on `anypriorprborfine`. Each unit change in `priorarrests=1` leads to a unit change in `anypriorprborfine`
24. DirectionalLinkage: Actions on `priorarrests≥5` will induce actions on `anypriorprborfine`. Each unit change in `priorarrests≥5` leads to a unit change in `anypriorprborfine`
25. DirectionalLinkage: Actions on `priorarrests≥2` will induce actions on ['priorarrestsforfelony']. Each unit change in `priorarrests≥2` leads to a unit change in `priorarrestsforfelony`
26. DirectionalLinkage: Actions on `priorarrests=1` will induce actions on ['priorarrestsforfelony']. Each unit change in `priorarrests=1` leads to a unit change in `priorarrestsforfelony`
27. DirectionalLinkage: Actions on `priorarrests≥5` will induce actions on ['priorarrestsforfelony']. Each unit change in `priorarrests≥5` leads to a unit change in `priorarrestsforfelony`
28. ReachabilityConstraint: The values of [`priorarrests≥2`, `priorarrests=1`, `nopriorarrests`, `priorarrests≥5`] must belong to one of 4 values with custom reachability conditions.
29. ReachabilityConstraint: The values of [`drugabuser`, `drugtreatment`] must belong to one of 4 values with custom reachability conditions.
30. ReachabilityConstraint: The values of [`alcoholabuser`, `alcoholtreatment`] must belong to one of 4 values with custom reachability conditions.

### C.1.3 Additional Results

This table includes additional model training and performance statistics. $p$ is the percent of positive points, $n$ is the number of points, $n_{clf\_pos}$ is the number of points that are classified as positive, and $n_{clf\_neg}$ is the number of points that are classified as negative.

|  | Train | Test |
|---|---|---|
| **AUC** | 0.704 | 0.702 |
| **Expected Calibration Error** | 0.19% | 0.24% |
| **Error** | 35.2% | 35.4% |
| $n$ | 15414 | 3854 |
| $n_{\text{pos}}$ | 7707 | 1927 |
| $p$ | 50.0% | 50.0% |
| $n_{\text{clf\_pos}}$ | 6407 | 1606 |
| $n_{\text{clf\_neg}}$ | 9007 | 2248 |

**Table 6:** Additional model statistics for the `recidivism` dataset

.

This figure is the `test` component of the left-most figure in Fig. 2.

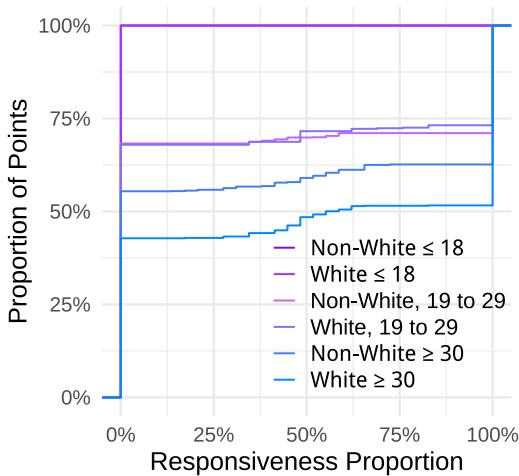

**Figure 5:** CDF of points by responsiveness percentage

**Ablation Testing** We performed additional ablation tests on the recidivism dataset, and show our results in the table below. We note that the pattern of unresponsiveness being higher among the non-white peisoners being higher than the white prisoners continues. The test AUC is also consistently lower when non-criminal history features (such as program participation and substance abuse) are removed.

| Dropped Features | Dropped Constraints | % Fixed (White) | | | % Fixed (Non-White) | | | AUC | |
|---|---|---|---|---|---|---|---|---|---|
| | | ≤ 18 | 19 - 29 | ≥ 30 | ≤ 18 | 19 - 29 | ≥ 30 | Train | Test |
| All age Bins | None | 39.0% | 65.6% | 39.0% | 51.4% | 69.2% | 51.4% | 0.696 | 0.686 |
| `drug_treatment` `alcohol_treatment` | 29, 30 | 49.8% | 66.8% | 49.8% | 61.9% | 73.3% | 61.9% | 0.699 | 0.69 |
| `drug_abuser` `alcohol_abuser` `drug_treatment` `alcohol_treatment` | 29, 30 | 50.5% | 67.3% | 50.5% | 61.8% | 73.1% | 61.8% | 0.698 | 0.691 |
| `edu_program_participation` `voc_program_participation` | None | 43.2% | 66.3% | 43.2% | 56.3% | 67.9% | 56.3% | 0.701 | 0.691 |
| None | All | 42.1% | 64.3% | 42.1% | 51.0% | 66.8% | 51.0% | 0.706 | 0.699 |

**Table 7:** Ablation testing results and details for each set of dropped features and constraints. Constraint numbers are from Table 5. $\epsilon$ and $\alpha$ are 0.1 and 0.05.

## C.2 Preventing Gaming in Content Moderation

### C.2.1 Description of Dataset

We work with the `twitterbot` which was originally curated by Gilani et al. [19]. The dataset defines a binary classification task where we wish to predict if an user account on Twitter belongs to a *human* ($y_i = 1$) or a *bot* ($y_i = 0$). The dataset contains a total of $n = 3{,}431$ instances and $d = 19$ features that encode semantically meaningful characteristics about their interactions and login history – e.g., `age_of_account_in_days` for account age, `user_tweeted` for the number of user tweets, and `source_identity` for source of user interaction (mobile, web, etc.).

In this case, the dataset contains a limited number of features given that all features are not readily available or shared across accounts. We process the dataset to define a subset of additional features as follows: (1) we include additional dummies to indicate "missing" values for `num_tweets`, `num_retweets` and `num_replies`; (2) we binarize features by using a adding a thermometer encoding to `num_followers` and `age_of_accounts_in_days`, setting thresholds that reflect salient milestones for follows and membership history; (3) we multi-hot encoded `source_identity`.

### C.2.2 Intervention Model

We consider an intervention model where each user can intervene on their platform interaction features. Our dataset contains 20 features of which 11 are actionable and 15 are mutable. Note that we do not allow interventions on features that a user cannot change themselves – i.e., number of followers.

We present a list of all features and their corresponding feature-level constraints in Table 8 and list joint actionability constraints below it.

**Exact Procedure**    We detail the intervention model for Exact procedure.

| Name | Type | LB | UB | Actionable | Sign | Joint Constraints | Partition ID |
|---|---|---|---|---|---|---|---|
| followers≥1k | $\{0,1\}$ | 0 | 1 | No | | 4 | 0 |
| followers≥100k | $\{0,1\}$ | 0 | 1 | No | | 4 | 0 |
| followers≥1M | $\{0,1\}$ | 0 | 1 | No | | 4 | 0 |
| followers≥10M | $\{0,1\}$ | 0 | 1 | No | | 4 | 0 |
| num_tweets | $\mathbb{Z}$ | 0 | 35000 | Yes | + | 1, 5 | 5 |
| no_tweets | $\{0,1\}$ | 0 | 1 | Yes | − | 1 | 5 |
| urls_count | $\mathbb{Z}$ | 0 | 13013 | No | | 5 | 5 |
| num_retweets | $\mathbb{Z}$ | 0 | 3000 | Yes | + | 2 | 6 |
| no_retweets | $\{0,1\}$ | 0 | 1 | Yes | − | 2 | 6 |
| num_replies | $\mathbb{Z}$ | 0 | 6991 | Yes | + | 3 | 7 |
| no_replies | $\{0,1\}$ | 0 | 1 | Yes | − | 3 | 7 |
| age_of_account≥180_days | $\{0,1\}$ | 0 | 1 | Yes | − | – | 1 |
| age_of_account≥365_days | $\{0,1\}$ | 0 | 1 | Yes | − | – | 2 |
| age_of_account≥730_days | $\{0,1\}$ | 0 | 1 | Yes | − | – | 3 |
| age_of_account≥1825_days | $\{0,1\}$ | 0 | 1 | Yes | − | – | 4 |
| follower_friend_ratio | $\mathbb{R}$ | 0.0 | 13364332.2 | Yes | − | – | 8 |
| source_web | $\{0,1\}$ | 0 | 1 | No | | – | 10 |
| source_mobile | $\{0,1\}$ | 0 | 1 | No | | – | 11 |
| source_app | $\{0,1\}$ | 0 | 1 | No | | – | 12 |
| source_news | $\{0,1\}$ | 0 | 1 | No | | – | 15 |

**Table 8:** Intervention Model for the processed `twitterbot` dataset. **Type** indicates the feature type ($\mathbb{Z}$ for integer, $\{0,1\}$ for binary). **LB**, **UB** are the lower and upper bounds for the feature. **Actionable** indicates whether the feature is globally actionable. Non-actionable features are highlighted. **Sign** indicates monotonicity constraints – whether feature can only increase (+) or decrease (-). **Joint Constraints** are a list non-separable constraint indices (listed below table) it is tied to (if any). **Partition ID** indicates which partition the feature belongs to.

1. IfThenConstraint: If notweets $= 0.0$, then numtweets $> 1.0$
2. IfThenConstraint: If noretweets $= 0.0$, then numretweets $> 1.0$
3. IfThenConstraint: If noreplies $= 0.0$, then numreplies $> 1.0$
4. DirectionalLinkage: Actions on numtweets will induce to actions on ['urlscount']. Each unit change in numtweets leads to at least 1.00-unit change in urlscount

 ## C.2.3 Additional Results

| Procedure | Description | Model Pool | | % Resp. (Perceived) | | | % Resp. (True) | | | AUC | | |
|---|---|---|---|---|---|---|---|---|---|---|---|---|
| | | # Models | # Cert. Robust | Train | Test | Valid | Train | Test | Valid | Train | Test | Valid |
| Manual | Train Models with Immutable Features | 370 | 370 | 0.0% | 0.0% | 0.0% | 0.0% | 0.0% | 0.0% | 0.531 | 0.570 | 0.581 |
| Convex | Consider Responsiveness w.r.t Convex Perturbation Check | 901 | 687 | 0.3% | 0.0% | 0.9% | 56.2% | 57.1% | 55.9% | 0.743 | 0.754 | 0.759 |
| Exact | Evaluate Responsiveness w.r.t Exact Actions | 901 | 76 | 9.6% | 9.9% | 9.3% | 9.6% | 9.9% | 9.3% | 0.722 | 0.727 | 0.734 |

**Table 9:** Full train, test, validation set results for the model with the highest validation AUC among *Considered* models: $\leq 10\%$ "Bot" predictions with certified responsiveness $\geq \varepsilon = 0.05$. *% Responsive* show % of "Bot" predictions with responsiveness $\geq \varepsilon = 0.05$ under the procedure's reachable set (*Perceived*) and the exact reachable set (*True*).

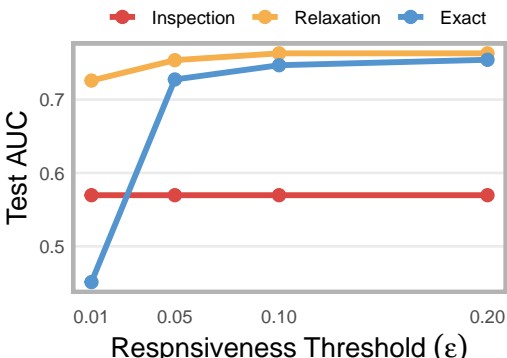

**Figure 6:** Test AUC of the best model that has less than 10% "Bot" predictions that have higher responsiveness than $\varepsilon = 0.01, 0.05, 0.1, 0.2$ for each procedure. Model does not change for Inspection since features are immutable.

## C.3 Organic Transplant Score

### C.3.1 Description of Dataset

As the availability of healthcare data is scarce and tightly regulated, we follow the methodology in the high-profile study of Attia et al. [3], who have demonstrated lower rates of prioritization for cancer patients using a simulated cohort of patients. Attia et al. generated the realistic simulated cohort by hand-crafting the probabilistic data model, and checking the resulting distributional characteristic against the real cohort of liver transplant patients. For this case study, we aim to reproduce their approach and derive a synthetic dataset which attains similar statistical properties. Specifically, we generate $n = 1,000$ simulated patients with $d = 32$ features.

We note that the TBS model itself is publicly available. For the purposes of our simulation, we reproduce its implementation based on an interactive R interface by Ewen Harrison.[1]

We use the default patient case from this implementation to set the baseline characteristics in our cohort. We modify certain variables in the default case as follows.

**Static Variables**   We simulate the demographics as follows:

$$\text{Age} \sim \text{Uniform}(\{30, 31, \ldots, 80\}) \tag{7}$$

$$\text{Gender} \sim \text{Bernoulli}(0.5) \quad \text{where } 0 = \text{man}, 1 = \text{woman}. \tag{8}$$

Note that we do not aim to have a representative distribution of a demographics in a population.

We also simulate other lab values as follows:

$$\text{Albumin} \sim \text{Uniform}[30, 40] \tag{9}$$

$$\text{Potassium} \sim \text{Uniform}[3.5, 5.0] \tag{10}$$

We detail the model for sampling other clinical variables next.

**Liver Parameters**   We set up the following structural causal model (SCM) [47] for the liver parameters: bilirubin, sodium, international normalized ratio (INR), and creatinine. We use this probabilistic model both to generate the initial patient cohort, and to simulate the random effects due to natural variation in the reachable sets.

Let $\mathbf{U} = (U_{\text{bili}}, U_{\text{Na}}, U_{\text{INR}}, U_{\text{creat}})$ denote the vector of exogenous noise variables, and let $x = (X_{\text{bili}}, X_{\text{Na}}, X_{\text{INR}}, X_{\text{creat}})$ denote the vector of correlated endogenous variables representing the four liver parameters.

*Exogenous Variables.* We set $\mathbf{U} \sim \mathcal{N}(0, \boldsymbol{\Sigma})$ with:

$$\boldsymbol{\Sigma} = \begin{bmatrix} 1.0 & 0.447 & 0.320 & -0.257 \\ 0.447 & 1.0 & 0.370 & -0.043 \\ 0.320 & 0.370 & 1.0 & -0.091 \\ -0.257 & -0.043 & -0.091 & 1.0 \end{bmatrix} \tag{11}$$

*Structural Equations.* The endogenous variables are determined by the following structural equations:

$$X_{\text{bili}} = \min(200, \max(15, \exp(0.5 \cdot U_{\text{bili}} + 3.5))) \tag{12}$$

$$X_{\text{Na}} = \min(145, \max(125, 5 \cdot U_{\text{Na}} + 137)) \tag{13}$$

$$X_{\text{INR}} = \min(2.4, \max(0.9, \exp(0.3 \cdot U_{\text{INR}} - 0.2) + 0.8)) \tag{14}$$

$$X_{\text{creat}} = \min(200, \max(45, \exp(0.4 \cdot U_{\text{creat}} + 4.2))) \tag{15}$$

where:

- $X_{\text{bili}}$ represents bilirubin levels (clipped to $[15, 200]$)
- $X_{\text{Na}}$ represents sodium levels (clipped to $[125, 145]$)
- $X_{\text{INR}}$ represents international normalized ratio (clipped to $[0.9, 2.4]$)
- $X_{\text{creat}}$ represents creatinine levels (clipped to $[45, 200]$)

We choose the parameters to approximately match the reported statistics in a simulated cohort from Attia et al. [3]. We show the statistical properties of our generated cohort in Fig. 7.

---

[1]https://github.com/SurgicalInformatics/transplantbenefit/

### C.3.2 Intervention Model

We detail the dataset features and the considered intervention model in the table:

| Name | Type | LB | UB | Actionable | Sign | Joint Constraints | Partition ID |
|---|---|---|---|---|---|---|---|
| rinpatient_tbs | $\{0,1\}$ | 0 | 1 | No | | – | 3 |
| rregistration_tbs | $\mathbb{Z}$ | 1 | 7 | No | | – | 4 |
| rwaiting_time_tbs | $\mathbb{Z}$ | 0 | 3650 | No | | – | 5 |
| rage_tbs | $\mathbb{Z}$ | 30 | 80 | No | | – | 6 |
| rgender_tbs | $\{0,1\}$ | 0 | 1 | No | | – | 7 |
| rdisease_primary_tbs | $\mathbb{Z}$ | 1 | 9 | Yes | | 1, 2 | 1 |
| rdisease_secondary_tbs | $\mathbb{Z}$ | 1 | 9 | No | | – | 8 |
| rdisease_tertiary_tbs | $\mathbb{Z}$ | 1 | 9 | No | | – | 9 |
| previous_tx_tbs | $\{0,1\}$ | 0 | 1 | No | | – | 10 |
| rprevious_surgery_tbs | $\{0,1\}$ | 0 | 1 | No | | – | 11 |
| rbilirubin_tbs | $\mathbb{Z}$ | 15 | 200 | No | | – | 2 |
| rinr_tbs | $\mathbb{R}$ | 0.9 | 2.4 | No | | – | 2 |
| rcreatinine_tbs | $\mathbb{Z}$ | 45 | 200 | No | | – | 2 |
| rrenal_tbs | $\{0,1\}$ | 0 | 1 | No | | – | 12 |
| rsodium_tbs | $\mathbb{Z}$ | 125 | 145 | No | | – | 2 |
| rpotassium_tbs | $\mathbb{R}$ | 3.5 | 5.0 | No | | – | 13 |
| ralbumin_tbs | $\mathbb{Z}$ | 30 | 40 | No | | – | 14 |
| rencephalopathy_tbs | $\{0,1\}$ | 0 | 1 | No | | – | 15 |
| rascites_tbs | $\{0,1\}$ | 0 | 1 | No | | – | 16 |
| rdiabetes_tbs | $\{0,1\}$ | 0 | 1 | No | | – | 17 |
| rmax_afp_tbs | $\mathbb{Z}$ | 0 | 1000 | No | | – | 18 |
| rtumour_number_tbs | {'0 or 1'*, '2', '3+'} | | | Yes | | 1, 2 | 1 |
| rmax_tumour_size_tbs | $\mathbb{R}$ | 0 | 20 | Yes | | 1, 2 | 1 |
| dage_tbs | $\mathbb{Z}$ | 18 | 80 | No | | – | 19 |
| dcause_tbs | $\mathbb{Z}$ | 1 | 4 | No | | – | 20 |
| dbmi_tbs | $\mathbb{R}$ | 15 | 50 | No | | – | 21 |
| ddiabetes_tbs | $\mathbb{Z}$ | 1 | 3 | No | | – | 22 |
| dtype_tbs | $\{0,1\}$ | 0 | 1 | No | | – | 23 |
| bloodgroup_compatible_tbs | $\{0,1\}$ | 0 | 1 | No | | – | 24 |
| splittable_tbs | $\{0,1\}$ | 0 | 1 | No | | – | 25 |

\* This feature value is treated as no tumours if the primary disease does not indicate cancer, rdisease_primary_tbs $\neq$ 1, and as one tumour otherwise.

1. IfThenConstraint: If rtumour_number_tbs $\in \{$'2','3+'$\}$, then rdisease_primary_tbs = 1 (cancer)
2. IfThenConstraint: If rdisease_primary_tbs = 1, then rmax_tumour_size_tbs > 0.

Concretely, to generate counterfactual patients with cancer, we define two intervention sets for small and large tumours, following Attia et al. [3]. In the small intervention set, we consider interventions so that the rtumour_number_tbs = '2' and rmax_tumour_size_tbs = 2; in the large intervention set, the number of tumours is the same but rmax_tumour_size_tbs = 5

**Random Effects** For generating noise around existing parameter values $\boldsymbol{x}^{(0)} = (x_{\text{bili}}^{(0)}, x_{\text{Na}}^{(0)}, x_{\text{INR}}^{(0)}, x_{\text{creat}}^{(0)})$, we first perform approximate abduction to infer the corresponding exogenous values $\mathbf{U}^{(0)}$ using the inverse structural equations. Then, we generate the perturbed exogenous variables as:

$$\mathbf{U}^{(1)} = \mathbf{U}^{(0)} + \boldsymbol{\varepsilon} \tag{16}$$

where $\boldsymbol{\varepsilon} \sim \mathcal{N}(\mathbf{0}, \boldsymbol{\Sigma})$ represents correlated noise. The counterfactual endogenous variables $\boldsymbol{x}^{(1)}$ are then computed by applying the structural equations to $\mathbf{U}^{(1)}$.

Thus, response probability distribution $\mathbb{P}_{\mathbf{a}}(\boldsymbol{x})$ is the distribution of $\Pr(\boldsymbol{x}^{(1)} - \boldsymbol{x}^{(0)} - \mathbf{a})$, where $\mathbf{a} = (a_{\text{bili}}, a_{\text{Na}}, a_{\text{INR}}, a_{\text{creat}})$ is the intervention.

### C.3.3 Additional Results

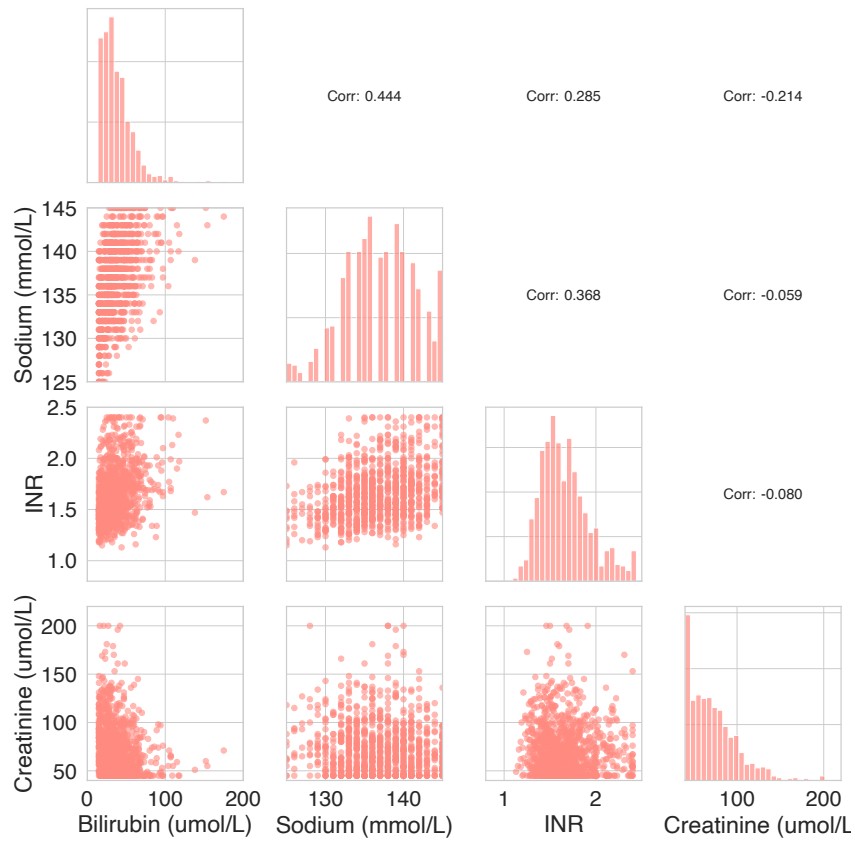

**Figure 7:** Pairwise relationships of the four liver parameter distributions according to our probabilistic model. These statistics are similar to those obtained by Attia et al. [3].

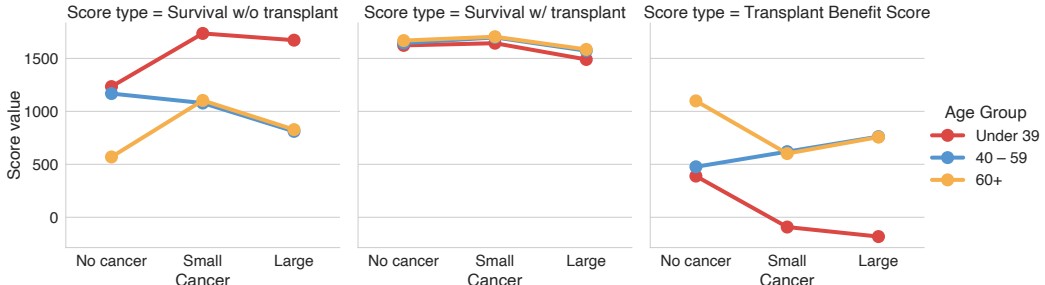

**Figure 8:** Average predictions of the TBS model and its components (*need model* on the left, *utility model* in the middle, combined on the right) over the reachable sets in the simulated cohorts. We can see that only for the middle-age group the average predicted survival w/o transplant decreases under the intervention, with other groups having the monotonicity constraints violated.

# D  Omitted Formal Results

**Remark 8.** *Given the $H_0$ and $H_1$ in Proposition 5 with confidence parameter $\alpha \in (0,1)$,*

$$\text{Reject } H_0 \implies n > \log \alpha / \log(1-\varepsilon)$$

**Proof.** *From the definition of the exact Binomial confidence interval, we have that:*

$$\rho_{2\alpha}^U(n, \hat{\rho}_n) = \mathsf{B}_{1-\alpha}(n\hat{\rho}_n + 1,\ n - n\hat{\rho}_n) \tag{17}$$

*provides a one-sided guarantee $\Pr(\rho(\boldsymbol{x}) \le \rho_{2\alpha}^U(n, \hat{\rho}_n)) \ge 1 - \alpha$.*

*The cumulative distribution of the Beta distribution is given by:*

$$\mathrm{F}(x; a, b) = \frac{\mathrm{B}(x; a, b)}{\mathrm{B}(a, b)}$$

*where $\mathrm{B}(x; a, b)$ is the incomplete beta function, defined as:*

$$\mathrm{B}(x; a, b) = \int_0^x t^{a-1}(1-t)^{(b-1)}\ dt$$

*and $\mathrm{B}(a, b) = \mathrm{B}(1; a, b)$.*

*Suppose $\hat{\rho}(\boldsymbol{x}) = 0$. Then our parameters for the beta distribution are $a = 1$, $b = n$. Hence,*

$$\mathrm{F}(x; 1, n) = 1 - (1 - x)^n$$

*Since the quantile function is the inverse of the CDF, we have*

$$\mathsf{B}_{1-\alpha}(1, n) = 1 - \alpha^{\frac{1}{n}}$$

*To reject $H_0$, we need $\mathsf{B}_{1-\alpha}(1, n) = 1 - \alpha^{\frac{1}{n}} < \varepsilon$. By rearranging the inequality, we have*

$$n > \frac{\ln(\alpha)}{\ln(1-\varepsilon)}$$

