# OpenReview forum: "Reliable Models via Responsiveness Verification"
_NeurIPS.cc/2025/Workshop/Reliable_ML — NeurIPS 2025 - Reliable ML Workshop_

### Official Review · Reviewer_usgx · 2025-09-17
**Interesting work with real world applications**

**Rating:** 7
**Confidence:** 2

**Review:**

### Summary:

The paper proposes a validation procedure for the responsiveness of predictions with respect to interventions on their features. Their basic assumption is that the features are semantically meaningful. They assume that each intervention has also downstream effects, which is realistic. They construct these estimates by generating a uniform sample of reachable points.

### Strengths:

S1. The assumptions made are realistic and justifiable. The assumption of downstream effects is very important for real-world applicability.

S2. The paper presents an interesting experimental evaluation in real-world settings.

### Weaknesses:

W1. The reachable points which belong to the reachable area where $||\alpha||_p \leq \delta$, there is only a very brief discussion on $\delta$ in lines 91-100. This should be expanded, and additional clarification should be provided regarding the distance metric, which might differ for each feature (e.g., different costs or recourse costs across features). This is a known and very important issue in the counterfactual explanations literature, and it should at least be mentioned.

W2. There is a strong dependence on domain knowledge, which makes it difficult to use in practice, especially for the downstream effects.

### Suggestions:

- Change one of the symbols ($\alpha$ or $a$), since they may cause confusion.

- Explicitly refer to Table 1 in the text.

- (Super minor:) Replace “we” in line 276.

---

### Official Review · Reviewer_jJsq · 2025-09-19
**A Complement to Actionable Recourse**

**Rating:** 7
**Confidence:** 2

**Review:**

he framework of sampling reachable points and estimating responsiveness with statistical guarantees is well-motivated, and the case studies in recidivism, organ transplant prioritization, and content moderation are compelling. My main concern is clarity on how “responsiveness” differs from the established notion of “actionable recourse.” Both operate on the same intervention space, but recourse focuses on producing feasible individual-level changes, while responsiveness evaluates the prevalence of such changes across a population as an audit tool. This distinction is present in the text but not emphasized strongly, and I encourage the authors to make it explicit, as readers familiar with recourse might otherwise see responsiveness as a rebranding. With that clarification, this work would be a good contribution to Reliable ML.